# Reward Design for Justifiable Sequential Decision-Making

**Aleksa Sukovic** [1,2]  **Goran Radanovic** [1]

Max Planck Institute for Software Systems [1]   Saarland University [2]

`{asukovic, gradanovic}@mpi-sws.org`

## Abstract

Equipping agents with the capacity to justify made decisions using supporting evidence represents a cornerstone of accountable decision-making. Furthermore, ensuring that justifications are in line with human expectations and societal norms is vital, especially in high-stakes situations such as healthcare. In this work, we propose the use of a debate-based reward model for reinforcement learning agents, where the outcome of a zero-sum debate game quantifies the justifiability of a decision in a particular state. This reward model is then used to train a justifiable policy, whose decisions can be more easily corroborated with supporting evidence. In the debate game, two argumentative agents take turns providing supporting evidence for two competing decisions. Given the proposed evidence, a proxy of a human judge evaluates which decision is better justified. We demonstrate the potential of our approach in learning policies for prescribing and justifying treatment decisions of septic patients. We show that augmenting the reward with the feedback signal generated by the debate-based reward model yields policies highly favored by the judge when compared to the policy obtained solely from the environment rewards, while hardly sacrificing any performance. Moreover, in terms of the overall performance and justifiability of trained policies, the debate-based feedback is comparable to the feedback obtained from an ideal judge proxy that evaluates decisions using the full information encoded in the state. This suggests that the debate game outputs key information contained in states that is most relevant for evaluating decisions, which in turn substantiates the practicality of combining our approach with human-in-the-loop evaluations. Lastly, we showcase that agents trained via multi-agent debate learn to propose evidence that is resilient to refutations and closely aligns with human preferences.

## 1 Introduction

Reinforcement learning (RL) has been achieving impressive successes in a wide range of complex domains. However, specifying a reward function which incentivizes RL agents to exhibit a desired behavior remains difficult (Leike et al., 2018). Prior work proposes several approaches that address these difficulties (Kwon et al., 2023; Bahdanau et al., 2018; Jothimurugan et al., 2019), including those based on learning from pairwise preferences (Christiano et al., 2017). However, such reward models are not informative enough for agents to learn how to substantiate their decisions with supporting evidence – a key property needed for accountable decision-making (Bovens, 2007). Hence, we ask the following question: *How can we design rewards that incentivize agents to carry out a given task, but through decisions that can be justified?*

To answer this questions, we consider a setting in which an RL agent acts as a principal, influencing the state of the world, while a human agent acts as a verifier responsible for validating the justifiability of the decisions taken by the RL agent, based on the evidence provided. This scenario mirrors common real-world situations where accountability is critical, including healthcare scenarios where doctors are tasked with scrutinizing the validity of automated decisions. We recognize three important properties that the provided evidence should satisfy. First, it should reflect the underlying state of the world, i.e., include the key information based on which an action was taken. A naive solution is to provide the full state as evidence. However, this discards the fact that the human, as a suboptimal-decision maker, may not easily reason about the taken decision because the state might be too large

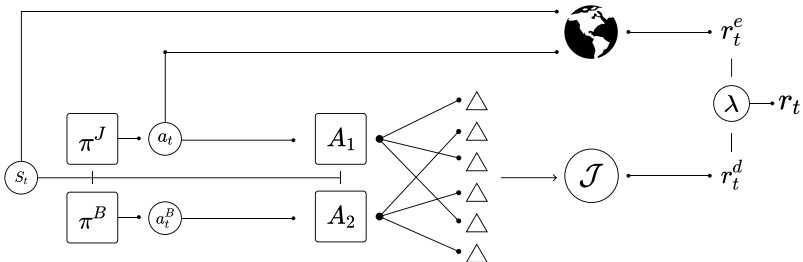

Figure 1: To obtain a debate reward $r_t^d$ in the state $s_t$, two argumentative agents $A_1$ and $A_2$ take turns proposing supporting evidence (depicted as triangles) for two decisions, up to a predefined limit (here, 3 evidence each). Then, a positive debate reward is issued whenever a proxy of a judge $\mathcal{J}$ considers action $a_t$, taken by the justifiable policy $\pi^J$, better justified than action $a_t^B$ taken by the baseline policy $\pi^B$. This reward is then mixed with the environment reward $r_t^e$ via debate coefficient $\lambda$, yielding the final reward $r_t$ used to train the justifiable agent.

or otherwise incomprehensible. This brings us to the second property: the provided evidence should also be concise and only reflect the most relevant information. The third property builds on the second one: given that the provided evidence contains only partial information about the state, this information should not be easy to refute. More specifically, additional information about the underlying state, e.g., additional evidence, should not change the human's judgment. Therefore, the overall challenge is to design a framework which can enable such justifications through a reward model that incentivizes both performance and justifiability.

To tackle this challenge, we consider a framework which modifies the environment rewards by mixing them with rewards coming from a debate-based reward model (see Figure 1). Each debate-based reward is defined through the outcome of a two-player zero-sum debate game. More specifically, we let two *argumentative* agents debate by taking turns providing supporting evidence contingent on the current state, each corroborating a decision made by one of two competing policies. Based on the proposed set of evidence, the human then states their preference over these two decisions, and this preference defines the *debate reward*. In this setup, one decision comes from a *baseline* policy, while the other comes from a *justifiable* policy that we optimize. Our approach builds upon the work of Irving et al. (2018), but we consider sequential decision-making problems and utilize debate to quantify a decision's justifiability. To this end, we recognize two main technical challenges. First, learning a proxy of a human judge that evaluates decisions solely based on the proposed evidence, with comparable performance to methods requiring full state exposure. Second, learning a representation of argumentative strategies that are solutions to different instances of the debate game. These two components are needed to enable efficient learning of the justifiable policy.

**Contributions.** Our contributions are as follows. (i) We formalize the problem of justifiable decision-making, modeling debate as an extensive-form game (Sec. 3). (ii) We provide a method for learning a proxy of a human judge that evaluates a decision's justifiability using proposed evidence (Sec. 4.2). (iii) We propose an approach to learning contextualized argumentative strategies that constitute approximate solutions of the debate games (Sec. 4.3). (iv) We conduct extensive empirical evaluation of our approach on a real-world problem of treating sepsis, testing the performance and justifiability of policies trained through our framework (Sec. 5.2), as well as the effectiveness and robustness of argumentative agents (Sec. 5.3, Sec. 5.4, and Sec. 5.5) [1].

## 2   RELATED WORK

**Debate.** Irving et al. (2018) first outlined theoretical and practical implications of debate as a method for training aligned agents. Debate has also been used to improve factuality of large language models (Du et al., 2023), reason in knowledge graphs (Hildebrandt et al., 2020), and aid in Q&A tasks (Perez et al., 2019). We build on this line of work by bringing the utility of debate in sequential decision-making problems, where it is used as an approach of quantifying justifiability of made decisions. In addition, by leveraging learning from pairwise preferences as in Christiano et al. (2017), our

---

[1]Our code is publicly available at github.com/aleksa-sukovic/iclr2024-reward-design-for-justifiable-rl.

framework enables encoding of human judgments in a form of the preferred evidence that renders a decision justified.

**Reward Design.** Previous work proposes several approaches that address difficulties of reward design, based on natural language (Kwon et al., 2023; Bahdanau et al., 2018), rule-based methods (Jothimurugan et al., 2019) and preferences (Biyik & Sadigh, 2018; Christiano et al., 2017). Furthermore, there is a line of work considering interpretable reward design, including reward sparsification (Devidze et al., 2021) and reward decomposition (Juozapaitis et al., 2019; Bica et al., 2020). Differently, we define a reward model as an outcome of a zero-sum debate game, which in itself is interpretable, and learn contextualized policies that solve it. Most similar to our methodology is learning from pairwise preferences as in Christiano et al. (2017), only we assume comparisons are made over justifying evidence, without exposure to the underlying state or trajectory.

**Explainable AI and Other Related Work.** Our approach is most similar to the attribution-based techniques for explaining the inner workings of models. In such approaches, contributions of input features are quantified numerically (Lundberg & Lee, 2017; Ribeiro et al., 2016) or visually represented as heatmaps (Selvaraju et al., 2017; Mundhenk et al., 2019). This line of research has received a significant attention, also in the context of explaining decisions of RL agents (Ragodos et al., 2022; Bastani et al., 2018). However, one limitation of these explanations is the inability to further align them with human preferences (Hadfield, 2021; Brundage et al., 2020). In contrast, our approach additionally enables human-specified justifications, facilitating the incorporation of preferences into their generation. There is also a line of work that examines approaches for adaptation of agent's recommendations (actions) to a baseline human policy in an expert-in-loop setup (Grand-Clément & Pauphilet, 2022; Faros et al., 2023). Differently, we consider a problem of learning to take actions that can be justified, where the agent itself acts as a primary decision-maker in the environment.

## 3 FORMAL SETUP

We consider a sequential decision-making problem, where an agent interacts with an environment over a series of time-steps, modeled by a discrete-time Markov Decision Processes (MDP). The episode begins by sampling a starting state from the initial state distribution $\rho$. At each time-step $t$, an agent observes the current state $s_t \in \mathcal{S}$, takes an action $a_t \in \mathcal{A}$ and receives an environment reward $r^e(s_t, a_t)$. The environment then transitions to a successor state $s_{t+1}$ with a probability specified by the transition function $T(s_{t+1}|s_t, a_t)$.

### 3.1 AGENTS

We consider two kinds of agents that operate in the defined MDP: *baseline* and *justifiable* agent.

**Baseline Agent.** The *baseline agent* aims to maximize the expected discounted value of the environment's return given as $\mathcal{R} = \sum_{t=0}^{T-1} \gamma^t r^e(s_t, a_t)$, where $\gamma \in [0, 1]$ is a user-specified discount factor. Its optimal action-value function, defined as $Q^*(s, a) = \max_\pi \mathbb{E}[\mathcal{R}|s_t = s, a_t = a, \pi]$, is the maximum expected discounted return obtained after taking action $a$ in state $s$. We will refer to a deterministic policy that maximizes the expected value of $\mathcal{R}$ as the *baseline policy* $\pi^B$, which satisfies $\pi^B(s) \in \text{argmax}_a Q^*(s, a)$. [2]

**Justifiable Agent.** While the baseline agent learns to solve the environment, its decisions may not be easy to justify when evaluated by a human. To design an agent that accounts for the justifiability of its decisions, we consider a reward defined as a weighted combination of the environment reward $r^e$ and the *debate reward* $r^d$, which encapsulates human judgment of justifiability and is specified in the next subsection. The expected return is then defined as:

$$\mathcal{R}_J = \sum_{t=0}^{T-1} \gamma^t \left[ (1 - \lambda) \cdot r^e(s_t, a_t) + \lambda \cdot r^d(s_t, a_t, a_t^B) \right]$$

where $\lambda \in [0, 1]$ is a *debate coefficient*, balancing between environment and debate rewards, and $a_t^B$ is the action of the baseline agent in state $s_t$. The value of $\lambda = 0.0$ corresponds to the return of the baseline policy, whereas for the value $\lambda = 1.0$ we obtain a setup similar to Christiano et al. (2017), where the agent relies only on the debate reward model. We refer to an agent that maximizes the expected value of $\mathcal{R}_J$ as the *justifiable agent*, denote its optimal action-value function as

---

[2]In practice, we require the baseline policy to be well-performing, but not necessarily optimal.

$Q_J^*(s,a) = \max_\pi \mathbb{E}\left[\mathcal{R}_J | s_t = s, a_t = a, \pi\right]$ and its deterministic policy $\pi^J(s) \in \mathrm{argmax}_a Q_J^*(s,a)$ as the *justifiable policy*.

## 3.2 Reward Modeling via Debate

Our objective is to learn a reward model, denoted as $r^d(s_t, a_t, a_t^B)$, which quantifies the justifiability of a decision. To this end, we introduce a reward model based on a *debate game*. In this model, $r^d(s_t, a_t, a_t^B)$ represents the value of a debate game induced by a tuple $(s_t, a_t, a_t^B)$. In more details, for a given tuple $(s_t, a_t, a_t^B)$, the induced debate game is formulated as a two-player zero-sum extensive-form game (Shoham & Leyton-Brown, 2008), in which the first player argues for taking the decision $a_t$ in state $s_t$, while the second player argues for taking the baseline decision $a_t^B$ in $s_t$.

**Debate Game.** With $\mathcal{N}$ we denote a set of nodes in a finite, rooted game tree. The action space is represented by a finite set of evidence $\mathcal{E}$, and each node $n \in \mathcal{N}$ consists of evidence proposed thus far in the game $n = \{e\} \subseteq \mathcal{E}$. Additionally, the edges to successors of each node define actions (evidence) $\{e : e \in \mathcal{E} \setminus n\}$ available to the acting player, where we disallow evidence repetition. The debate game is a perfect-information game: at all times, the players have no ambiguity about the evidence proposed up until the current point and have a complete knowledge about the state of the game. The game proceeds as players take turns: in turn $l$, player $i = l \mod 2 + 1$ proposes evidence $e_{l/2}^i$.[3] The total number of turns $L$ is assumed to be even and significantly smaller than the evidence set, i.e., $L \ll |\mathcal{E}|$. After the last turn, a terminal node $n_L = (e_1^1, e_1^2, ..., e_{L/2}^1, e_{L/2}^2) = \{e_{n_L}\}$ is evaluated. The players' utilities are $u_1(n_L) = -u_2(n_L) = \mathbb{U}(a_t, a_t^B, \{e_{n_L}\})$, with $\mathbb{U}$ defined as:

$$\mathbb{U}(a_t, a_t^B, \{e\}) = \begin{cases} +1, & \mathcal{J}(a_t, \{e\}) > \mathcal{J}(a_t^B, \{e\}) \\ 0, & \mathcal{J}(a_t, \{e\}) = \mathcal{J}(a_t^B, \{e\}) \\ -1, & \text{otherwise} \end{cases}$$

Here, $\mathcal{J}$ is a model of a human judge that, for a given decision $a$ and evidence $\{e\}$, outputs a numerical value $\mathcal{J}(a, \{e\}) \in \mathbb{R}$ quantifying how justifiable $a$ is under evidence $\{e\}$.

**Strategies and Solution Concept.** A player's strategy $\sigma^i : \mathcal{N} \rightarrow \mathcal{E}$ outputs available evidence $\sigma^i(n) \in \mathcal{E} \setminus n$ in a given node $n$ and $\Sigma^i$ is the set of all strategies of the player $i$. Based on the utility function, we additionally define $\mathcal{G}(\{\sigma^1, \sigma^2\}, s_t, a_t, a_t^B)$ as the *payoff* (utility) of the first player, conditioned on both players following the strategy profile $\{\sigma^1, \sigma^2\}$. Then, a set of the best responses of the first (resp. second) player to its opponent strategy $\sigma^2$ (resp. $\sigma^1$) is defined as $b^1(\sigma^2) = \mathrm{arg\,max}_{\sigma^1 \in \Sigma^1} \mathcal{G}(\{\sigma^1, \sigma^2\}, s_t, a_t, a_t^B)$ (resp. $b^2(\sigma^1) = \mathrm{arg\,min}_{\sigma^2 \in \Sigma^2} \mathcal{G}(\{\sigma^1, \sigma^2\}, s_t, a_t, a_t^B)$). A strategy profile $\bar{\sigma} = \{\bar{\sigma}^1, \bar{\sigma}^2\}$ is a pure-strategy Nash equilibrium if $\bar{\sigma}^i \in b^i(\bar{\sigma}^{-i})$. Because the debate game is a perfect-information extensive-form game, a pure-strategy Nash equilibrium exists (Shoham & Leyton-Brown, 2008), and due to its zero-sum structure, it can be obtained by solving the following max-min optimization problem: $\max_{\sigma^1 \in \Sigma^1} \min_{\sigma^2 \in \Sigma^2} \mathcal{G}(\{\sigma^1, \sigma^2\}, s_t, a_t, a_t^B)$. We refer to $\mathcal{G}(\{\bar{\sigma}^1, \bar{\sigma}^2\}, s_t, a_t, a_t^B)$ as the value of the game [4] and define $r^d(s_t, a_t, a_t^B)$ to be equal to it, i.e., $r^d(s_t, a_t, a_t^B) = \alpha \cdot \mathcal{G}(\{\bar{\sigma}^1, \bar{\sigma}^2\}, s_t, a_t, a_t^B)$, where $\alpha > 0$ is a scaling coefficient.

## 4 Learning Framework

To effectively use a debate game outcome during training of justifiable policies, it is necessary to devise a model of a human judge $\mathcal{J}$ (Sec. 4.2) that encapsulates justifiability judgment and additionally learn argumentative policies that approximate a Nash equilibrium and are able to generalize across different instances of the debate game. With $\hat{r}^d(s_t, a_t, a_t^B)$ we denote an approximation of $r^d(s_t, a_t, a_t^B)$, obtained by running the argumentative policies from Sec. 4.3 in the debate game induced by $(s_t, a_t, a_t^B)$. In all our experiments, we set $\alpha = 5$.

---

[3]In our implementation of the debate game, we randomly chose which player has the first turn, i.e., $i = (l + \tau) \mod 2 + 1$, where $\tau \sim \mathcal{U}(\{0, 1\})$. This only affects the order of the evidence in $n_L$.

[4]For a two-player zero-sum game, the value of the game (or payoff) is unique (von Neumann & Morgenstern, 1947).

### 4.1 PREFERENCE DATASET

We assume the human judgments are collected in a preference dataset $\mathcal{D}$ of tuples $(s_t, a_0, a_1, p)$, where $s_t$ is a state, $a_0 \neq a_1$ are two decisions and $p \in \{0, 1\}$ indicates which of the two decision is more justified in a particular state. The value of $p = 0$ (resp. $p = 1$) indicates that $a_0$ (resp. $a_1$) is more preferred.

### 4.2 JUDGE MODEL

Because asking for human feedback is expensive, we aim to learn a judge model from the dataset $\mathcal{D}$ of preferences that can be used to evaluate the outcome of debate games. The judge, parametrized with learnable parameter $\theta \in \mathbb{R}^{d_1}$, is defined as a scalar reward function $\mathcal{J}_\theta(a, \{e\}) \in \mathbb{R}$ quantifying how justifiable a decision $a$ is, given evidence $\{e\}$. Similar to Christiano et al. (2017), we additionally assume that justifiability preference for decision $a_0$ over decision $a_1$ follows the Bradley-Terry model (Bradley & Terry, 1952):

$$\mathcal{P}(a_0 \succ a_1, \{e\}) = \frac{\exp \mathcal{J}_\theta(a_0, \{e\})}{\exp \mathcal{J}_\theta(a_0, \{e\}) + \exp \mathcal{J}_\theta(a_1, \{e\})}.$$

Here, we require the judge to quantify the level of justifiability given all evidence at once. Note the lack of dependency on the state $s_t$: the judge evaluates only proposed evidence, whereas the argumentative agents are in charge of providing those evidence, contingent on the state. We optimize the parameters by minimizing the cross-entropy loss between preference-predictions and labels from the dataset. See App. C.1 for more details.

### 4.3 ARGUMENTATIVE AGENT

Our overarching goal is to learn a generalizable argumentative policy that is able to solve any debate game, conditioned on its defining tuple $(s_t, a_t, a_t^B)$. This is a difficult feat, as the evidence set available to the agent is contingent on the state $s_t$. To this end, we can treat the debate game as an instance of a contextualized extensive form game (Sessa et al., 2020), where we consider the tuple $(s_t, a_t, a_t^B)$ as a context $z \in \mathcal{Z}$. In a general case, the justifiable agent $\pi^J$ observes the state $s_t$ and takes action $a_t$ which, paired with an action $a_t^B$ the baseline agent $\pi^B$ would have taken, sets the debate game context $z = \{s_t, a_t, a_t^B\}$. In the specific case of offline reinforcement learning we consider here, the contexts are sampled i.i.d. from the static dataset throughout training of both, argumentative and justifiable agents. Therefore, given a sample from the preference dataset $(s_t, a_0, a_1, p) \sim \mathcal{D}$, we set the context to $z = (s_t, a_p, a_{1-p})$. Player $i$'s contextual strategy, parametrized with learnable parameter $\phi_i \in \mathbb{R}^{d_2}$, is defined as $\sigma_{\phi_i}^c : \mathcal{Z} \to \Sigma^i$, mapping a context to the strategy of the player for the induced debate game [5]. To learn parameters $\phi_i$, we solve: $\max_{\phi_1 \in \mathbb{R}^{d_2}} \min_{\phi_2 \in \mathbb{R}^{d_2}} \mathbb{E}_{z \sim \mathcal{D}}[\mathcal{G}(\{\sigma_{\phi_1}^c(z), \sigma_{\phi_2}^c(z)\}, z)]$.

### 4.4 METHOD

**Judge.** The judge $\mathcal{J}_\theta$ is parametrized by weights $\theta \in \mathbb{R}^{d_1}$ of a neural network with two fully-connected layers of size 256, using parametric relu (He et al., 2015) activation and batch normalization (Ioffe & Szegedy, 2015). The network receives a vector $x$ in $\mathbb{R}^{|\mathcal{E}|}$, where only the values of evidence $\{e\}$ are shown, while the rest are set to zero. In addition, a binary mask of the same dimension, wherein all elements corresponding to the evidence $\{e\}$ are assigned a value of one, while the remaining elements are set to zero, as well as a one-hot encoded action $a$ are passed. The learning is done for a total of 100 epochs using batches of 64 comparisons sampled from the preference dataset $\mathcal{D}$, Adam optimizer and a learning rate of 5e-4. See App. C.1 for more details.

**Argumentative Agent.** We represent parameters $\phi \in \mathbb{R}^{d_2}$ of the argumentative agent $\sigma_\phi^c(\cdot|z)$, as weights of a neural network composed of 2 fully-connected layers of size 512 with leaky-relu activation function (Maas et al., 2013) with slope of 1e-2. The network takes as input a 44-dim state, a one-hot encoded decision for which the agent argues, as well as a binary mask of currently proposed evidence. The evidence (action) space of the policy is discrete and has 44 choices, each corresponding to exactly one state feature. To train the agent, we use PPO (Schulman et al., 2017)

---

[5]Equivalently, we will also denote the contextual strategy of player $i$ as $\sigma_{\phi_i}^c(\cdot|z) : \mathcal{N} \to \mathcal{E}$ for $z \in \mathcal{Z}$.

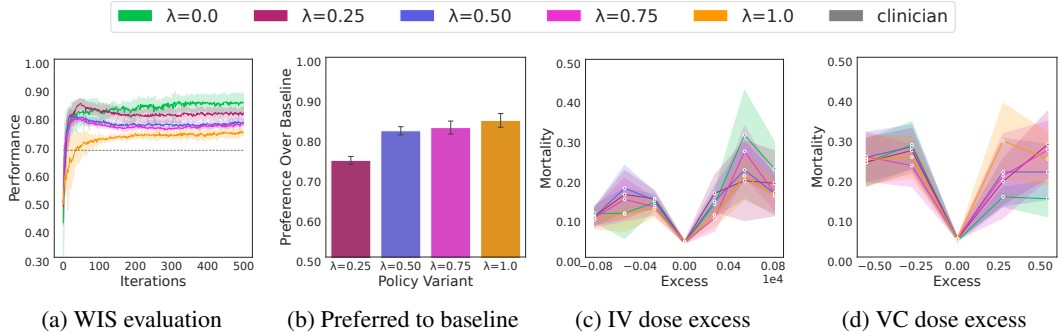

Figure 2: Evaluation of justifiable policies. For (b)-(d), the confidence intervals represent $\pm 2$ standard errors of the mean over 5 random seeds. (a) Policy performance as measured by WIS evaluation on a held-out test set with $\pm 1$ terminal rewards for every patient discharge or death. The mean and standard deviation are reported over 5 random seeds. (b) Percent of times judge preferred decisions of justifiable policies (i.e., $\lambda > 0.0$) compared to those of the baseline policy (i.e., $\lambda = 0.0$). (c) (d) Observed patient mortality (y-axis) against variations in IV/VC treatment doses prescribed by clinicians compared to the recommendations of learned policies (x-axis).

and examine two optimization strategies, namely *self-play* and *maxmin*. We train the *self-play* debate by letting the agent argue with a copy of itself [6], updating that copy every 100k steps, and repeating the procedure for 500 generations. For *maxmin* debate, we use 2 different set of weights, $\phi_1$ and $\phi_2$, to represent the agents' policies. The first agent i.e., $\phi_1$ is trained for 4k steps, followed by training of the second agent i.e., $\phi_2$ for 100k steps, repeating this procedure for 500 generations. This approach of overfitting the second agent to the current version of the first ensured learning of defense strategies against a very strong adversary. See App. C.2 for more details.

**Justifiable Agent.** To learn a justifiable policy $\pi^J(s) \in \arg\max_a Q_J^*(s, a)$, we consider model-free reinforcement learning methods based on Q-learning. Our approach builds on top of Raghu et al. (2017) and is based on a variant of Deep-Q networks (Mnih et al., 2015), specifically double-deep Q-network with the dueling architecture (Wang et al., 2016; Van Hasselt et al., 2016). The final network consists of 2 fully-connected layers of size 128 using leaky-relu activation function with slope 1e-2. The learning is done in batches of 256 $(s, a, r, s')$ tuples sampled from a Prioritized Experience Replay buffer (Schaul et al., 2015) using a learning rate of 1e-4, for a total of 25k iterations, evaluating the policy every 50 iterations on a held-out test set. When incorporating signal from the debate, i.e., $\hat{r}^d$, we augment the reward from the replay buffer as described in Sec. 3.1. Modifying just the observed reward, without changing the state dynamics, has been shown to be sufficient to induce learning of an alternative policy (Ma et al., 2019). See App. C.3 for more details, including additional loss terms and a list of all hyperparameters.

## 5 EXPERIMENTS

In our experiments, we aim to empirically evaluate the properties of debate as a method for reward specification. To this end, we perform quantitative and qualitative evaluation of justifiable policies (Sec. 5.2), analyze the effect of pairwise comparison only over proposed evidence on performance and justifiability (Sec. 5.3), examine the necessity of multi-agent debate in learning to provide evidence that is resilient to refutations (Sec. 5.4), and compare efficiency in providing supporting evidence for a decision of argumentative policies and SHAP feature-attribution method (Sec. 5.5).

### 5.1 ENVIRONMENTAL SETUP

**Sepsis.** Data for our cohort were obtained following steps outlined in Komorowski et al. (2018), utilizing MIMIC-III v1.4 database (Johnson et al., 2016). We focus our analysis on patients that fulfill Sepsi-3 criteria (Singer et al., 2016), 18,585 in total. The patient vector consists of 40 continuous and 4 discrete features, and the action space consists of $5 \times 5$ discrete choices of intravenous

---

[6]In terms of learnable parameters, this implies $\phi_2 \coloneqq \phi_1$.

(IV) fluids and vasopressors (VC). As in Raghu et al. (2017), the environment rewards are clinically guided. We set the environment reward $r^e$ to $\pm 15$ for terminal states resulting in patient survival or death, respectively, and shape the intermediate rewards based on the SOFA score (measure of organ failure). See App. B for more details.

**Preference Dataset.** To make a comprehensive evaluation possible, we define a synthetic ground-truth preference by making an assumption that a human judge always prefers a treatment prescribed by the clinician. Therefore, clinician's actions are the justified actions in our experiments. More formally, we bootstrap the dataset of preferences $\mathcal{D}$ by matching every pair $(s_t, a_t)$ from the cohort with an alternative action $a_r \sim \mathcal{U}(A)$, $a_r \neq a_t$ sampled uniform-random from the action space, initializing the preference variable $p$ to point to the true action $a_t$, as taken by the clinician (see App. D.5 for alternative dataset bootstrapping methods). The number of evidence is fixed to 6 ($\sim 13.6\%$ of the full state) (see App. D.1 for results with $L = 4$ evidence). The dataset is split into chunks of 70%, 15%, 15% used for training, validation, and testing respectively. We report all our results on a held-out test set, not seen by any of the agents nor the judge. When training a judge model, we additionally augment a tuple $(s_t, a_0, a_1, p)$ with an evidence set $\{e\}$. To generate it, we sample a predetermined number of state features uniform-random i.e., $\{e\} \sim \mathcal{U}(s_t)$, from the state, which is inspired by Irving et al. (2018) and ensures evidence is contingent on the state (Sec. 3.2) [7]. On the test set, the judge is able to correctly recover the underlying preference from the dataset with a relatively low accuracy of 65%.

**Baselines.** When comparing effectiveness of treatment policies, we consider two baselines. First, we consider the observed reward of the clinician from the dataset (depicted as a gray horizontal line in plots). Second, the baseline policy (Sec. 3.1) serves as an indicator of the optimal treatment policy. To demonstrate the robustness of multi-agent debate, we introduce an *isolated* argumentative agent. This agent aims to find an evidence set $\{e\}$ that maximizes $\mathcal{J}(a_p, \{e\})$, for a given $a_p$. To achieve this, we solve a search problem akin to the debate game by applying reinforcement learning (see App. C for more details). Lastly, we use SHAP when comparing effectiveness of debate to a feature-importance approach in providing supporting evidence for a decision.

## 5.2 EXPERIMENT 1: EFFECTIVENESS OF TASKS POLICIES

To examine the potential of specifying the reward as an adversarial game and its effect on the quality of learned behaviors, we train several policies by varying the debate coefficient $\lambda$.

**Quantitative Evaluation.** In Plot 2a, we evaluate the performance of different justifiable policies on a held-out test set during the course of training using weighted importance sampling (WIS) (Sutton & Barto, 2018) [8]. Likewise, in Plot 2b, we show the judge's preference over decisions made by justifiable policies (i.e., $\lambda > 0.0$), compared to the baseline policy (i.e., $\lambda = 0.0$), when the two were different [9]. The observed inherent trade-off between performance and justifiability suggests that tuning the debate coefficient $\lambda$ is important in practice, and we further elaborate on this in Sec. 6.

**Qualitative Evaluation.** In addition to quantitative evaluation, we perform qualitative analysis similar to Raghu et al. (2017). Plots 2c and 2d showcase correlation between observed mortality and difference between the optimal doses suggested by policies, and the actual doses prescribed by the clinicians. For all trained policies, the lowest mortality is observed when the difference is near zero, thus further showcasing their potential validity. It is also encouraging to see that the policy trained solely with debate rewards (i.e., $\lambda = 1.0$) remains quantitatively and qualitatively competitive in addition to being highly favored by the judge, even though it relies only on debate rewards.

## 5.3 EXPERIMENT 2: DEBATE-BASED FEEDBACK VS. STATE-BASED FEEDBACK

Because the judge is evaluating justifiability using only proposed evidence, a natural question to ask is how much does this affect the performance and alignment of trained policies. To provide an answer, we train a new judge that evaluates decisions based on the full state, namely $\mathcal{J}'(a_t, s_t)$. We then use this judge instead of $\hat{r}^d$ to train a new justifiable policy using *state-based* feedback, one

---

[7] We discuss different approaches for defining a set of evidence in Sec. 6.

[8] The behavior policy used in WIS was derived via behavior cloning, further described in App. C.3.

[9] See Plot 6a in App. D for a detailed breakdown of actions proposed by justifiable policies.

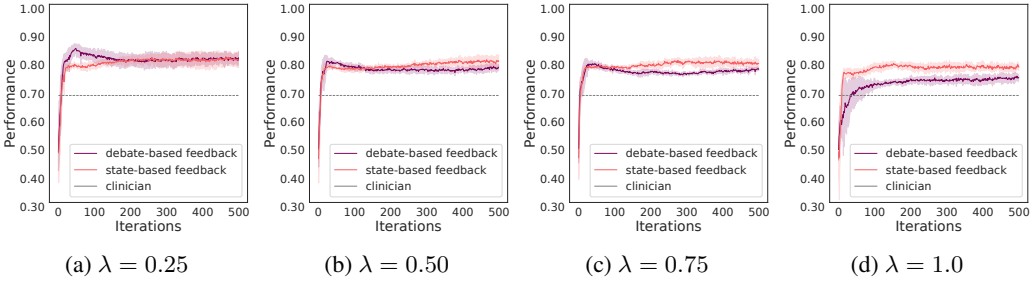

(a) $\lambda = 0.25$     (b) $\lambda = 0.50$     (c) $\lambda = 0.75$     (d) $\lambda = 1.0$

Figure 3: Performance of policies trained with state-based feedback compared to debate-based feedback, as measured by the weighted importance sampling evaluation on a held-out test set with $\pm 1$ terminal rewards for every patient discharge or death. The mean and standard deviation are reported over 5 random seeds.

for each $\lambda$. Given a sample $(s_t, a_0, a_1, p) \sim \mathcal{D}$, we consider a policy $\pi$ aligned to the ground-truth preference if $Q^*_\pi(s, a_p) \geq Q^*_\pi(s, a_{1-p})$.

The results are shown in Plot 4a for various policies trained with state- and debate-based feedback. We see that, even though the debate-based approach uses only $\sim 13.6\%$ of the full state, the achieved level of alignment remains similar. Likewise, in Plots 3a-3d we observe a matching trend when it comes to performance of the policies, although here we additionally note that the difference seems to increase in favor of policies trained with the state-based feedback as we increase the debate coefficient $\lambda$. These results seem promising, as they indicate one can expect to obtain a competitive level of alignment and performance, while requiring the judge to elicit preference over relatively small number of evidence.

## 5.4 EXPERIMENT 3: EFFECTIVENESS OF ARGUMENTATIVE POLICIES

**Preference Recovery Rate.** We recall the judge's accuracy in correctly predicting the preferred action $a_p$ from the preference dataset was $65\%$. To evaluate the effectiveness of argumentative policies, for each sample $(s_t, a_0, a_1, p) \sim \mathcal{D}$ we measure the judge's accuracy in predicting the more justified action $a_p$, when evidence $\{e\}$ is provided by one of the argumentative policies. In Plot 4b (green) we show the judge's accuracy when different argumentative agents propose $L = 6$ required evidence for action $a_p$, averaged across 1000 different debate games. The judge's accuracy is boosted from $65\%$ to a near $90\%$, demonstrating that agents can significantly amplify the capabilities of an otherwise limited judge.

**Robust Argumentation.** A good supporting evidence is both convincing and not easily refutable by counterarguments. To test the robustness of the proposed evidence, we train 3 adversarial *confuser* agents [10], each targeting one of the three (frozen) argumentative agents. The goal of the confuser is to convince the judge of the opposing action $a_{1-p}$. For $L = 6$, to obtain evidence $\{e\}$, the agent (maxmin or self-play) and its corresponding confuser take turns and propose 3 evidence each. The isolated agent is trained to first propose $L = 3$ evidence, followed by the confuser proposing the remainder (see App. C.2.1 for more details and App. D.3 for additional results). Plot 4b (red) shows judge's accuracy in this setting for 1000 different debate games. We observe that the isolated argumentative agent (Sec. 5.1) is not resilient to refutations, enabling the confuser to bring the judge's accuracy down to $38\%$, effectively convincing it of the opposite of its preference. Differently, both maxmin and self-play agents managed to keep the judge's accuracy to about $85\%$.

## 5.5 EXPERIMENT 4: COMPARISON TO SHAP-BASED EXPLANATIONS

A widely used approach to analyze black-box machine learning models is through feature-attribution techniques. We aim to demonstrate that these explanations may not necessarily be as effective when used as supporting evidence. We focus on the SHAP framework (Lundberg & Lee, 2017), specifically in providing justifications for decisions of various justifiable policies (Sec. 5.2). To justify a

---

[10]Confuser agents use the same architecture as argumentative agents, further described in App. C.

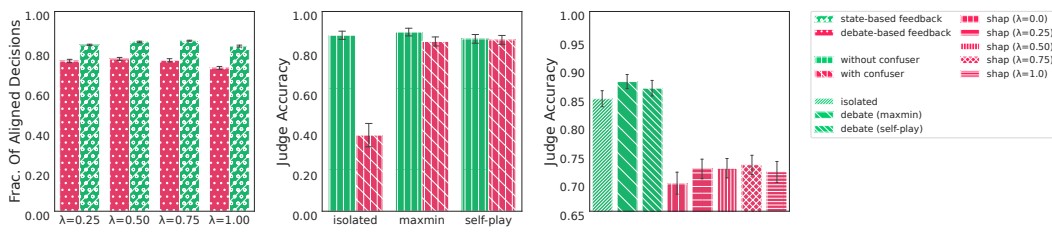

(a) State/debate feedback  (b) Preference recovery  (c) Comparison to SHAP

Figure 4: (a) Fraction of aligned decisions of policies trained with debate- and state-based feedback. Confidence intervals (CI) represent $\pm 2$ standard errors of the mean over $5$ random seeds. (b) Accuracy of the judge in predicting the preferred action, with and without the confuser agent, with CI representing $\pm 2$ standard errors of the mean estimate. (c) Effectiveness of SHAP-based explanations when used to justify a decision, as measured by the judge's accuracy, with CI representing $\pm 2$ standard errors of the mean estimate.

decision using SHAP, we select the top 6 state features, as ranked by their Shapely values. For argumentative models, we either run the debate between $a_0$ and $a_1$ (for maxmin and self-play agents) or propose 6 evidence in isolation (for isolated agent). In Plot 4c, we show the judge's accuracy against different approaches of proposing the evidence $\{e\}$, across 1000 comparisons $(s_t, a_0, a_1, p)$ sampled uniform-random from the test set. The SHAP-based evidence do improve the accuracy to about $70\%$, but nevertheless fall short compared to the argumentative agents.

# 6 DISCUSSION

In this work, we proposed use of a debate-based reward model as a general method of specifying desired behavior and necessary evidence to justify it. In this section, we take a step back and touch upon a couple of aspects that are relevant to future research based on this work.

**Eliciting Preferences.** The success of debate depends on the human's capability of judging its outcome, a process which may be affected by one's beliefs and biases. Extra care must be taken when collecting large datasets of preferences, as *belief bias* [11] is known to alter the results of judgment in human evaluations (Anderson & Hartzler, 2014) and is amplified in time-limited situations (Evans & Curtis-Holmes, 2005), which human annotators frequently encounter. Furthermore, in our experiments, we assumed existence of a single preference (clinician's decision). However, preferences collected from multiple raters will undoubtedly yield a higher variability in this respect. Motivated by positive results seen in this work, future research could undertake further studies that examine the effectiveness of eliciting preferences over partial state visibility through use of the debate as an amplification method.

**Defining Arguments is Difficult.** We have focused on debate assuming a well-defined argument space. While state features are one possible and easy choice, finding clear and expressive arguments presents a challenge, impacting the applicability of debate. In the context of RL, one potential alternative is considering previous trajectories in support of the current decision. In domains involving text generation, evidence could be defined as sentences, paragraphs, or references supporting a claim. A potentially interesting research direction is to examine the utility of a debate in a domain of human-ai collaboration, specifically in sequential decision-making tasks.

**Practical Considerations.** We would like to emphasize that when deploying our framework in practice, it is important to account for the context in which the system is being employed. The performance-justifiability trade-off from our experiments suggests that a special care ought to be given to selecting hyperparameters, in particular, those that weight the importance of the environment rewards and debate-based rewards. In practice, this means that a reward designer has to assess the potentially differing objectives encoded in these values as well as the accuracy of the proxy judge model, prior to the training process.

---

[11] A *belief bias* represents a tendency to judge the strength of arguments based on the plausibility of the conclusion, instead based on how strongly they support the conclusion.

## ETHICS STATEMENT

We acknowledge a potential misuse of the debate framework for malevolent purposes, such as deception. In the advent of AI systems that surpass human performance in many tasks, novel approaches must be developed which enable defense against malicious agents. While recent results indicate that debate is useful in thwarting malicious use of AI systems, further research is paramount in ensuring one can detect and defend against nefarious purposes. Moreover, the reliance on human judgments introduces the possibility of capturing their biases. Subsequently, designed reward function can incentivize argumentative agents to amplify those biases or learn to leverage them to achieve their objective. This is particularly important for practical considerations (Sec. 6), where a reward designer is tasked with assessing and handling the performance-justifiability trade-off. It is important to note that our results do not hint at solutions for dealing with these challenges; they only demonstrate that the trade-off exists. It is therefore of great importance that future research investigates novel algorithmic approaches and methods to tackle such challenges.

## ACKNOWLEDGMENTS

This research was, in part, funded by the Deutsche Forschungsgemeinschaft (DFG, German Research Foundation) – project number 467367360. We thank the anonymous reviewers for their valuable comments and suggestions.

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

## A    LIST OF APPENDICES

In this section, we provide a brief description of the content provided in the appendices of the paper.

- Appendix B provides details about the patient dataset and the defined environment:
    - o B.1 provides a general background on sepsis;
    - o B.2 provides details about the patient cohort;
    - o B.3 provides details about the environment's action space;
    - o B.4 provides details about the environment's reward structure.
- Appendix C provides details about the models:
    - o C.1 provides details about the judge model;
    - o C.2 provides details about the argumentative agents and their confusers;
    - o C.3 provides details about justifiable and baseline agents.
- Appendix D provides additional results:
    - o D.1 provides results involving debates with $L = 4$ evidence;
    - o D.2 provides further analysis of justifiable agent's actions;
    - o D.3 provides additional results pertaining to the isolated agent;
    - o D.4 provides results for agents using an alternative definition of the utility function;
    - o D.5 provides analysis of alternative methods for the preference dataset definition.

## B    DATASET

### B.1    SEPSIS

Sepsis is a life-threatening condition, defined as severe infection leading to an acute organ dysfunction, which in turn can cause a cascade of changes that damage multiple organ systems (Singer et al., 2016). It is also one of the leading causes of patient mortality (Cohen et al., 2015). Apart from administration of antibiotics and control of the infection source, a critical challenge in management of sepsis lies in the administration of intravenous fluids (IV) and vasopressors (VC). While international efforts attempt to provide a general guidance (Dellinger et al., 2004), clinicians are nevertheless tasked in devising individualized treatments based on specificities of patients.

A first step towards automated management of septic patients was done in a seminal work of Komorowski et al. (2018). The problem of treating sepsis was tackled by applying reinforcement learning to devise optimal treatment strategies for prescribing doses of intravenous fluids and vasopressors. The authors discretized the possible doses, which resulted in an action-space consisting of 25 distinct choices. The patient data was obtained from the MIMIC-III dataset (Johnson et al., 2016), and the authors extracted a subset of 48 patient features, discretized into 4h time windows. The preliminary results indicated that a learned policy was both quantitatively and qualitatively desirable. Since then, several other works extend and improve upon on this line of research. For example, Raghu et al. (2017) proposed a continuous state-space approach based on deep reinforcement learning, on which we build on in this work. Likewise, Huang et al. (2022) proposed an approach that is based on continuous action-space, thus enabling more granular control of prescribed doses.

### B.2    PATIENT COHORT

To assemble our patient cohort, we utilize the MIMIC-III v1.4 database (Johnson et al., 2016), focusing our analysis on patients that fulfill the Sepsis-3 criteria (Singer et al., 2016). Similar as in Komorowski et al. (2018), the sepsis is defined as a suspected existence of infection (indicated by the prescribed antibiotics) paired with a mild evidence of organ dysfunction (indicated by the

Table 1: List of features comprising the patient vector. We select 40 time-varying continuous features and 4 demographic discrete features.

Continuous features

| SOFA | Calcium | Shock index |
|---|---|---|
| Urine output 4h | Urine output total | Cumulated balance |
| Glasgow coma scale | Heart rate | Systolic blood pressure |
| Mean blood pressure | Diastolic blood pressure | Total input fluids |
| Respiratory rate | Temperature | FiO2 - fraction of inspired oxygen |
| Potassium | Sodium | Chloride |
| Glucose | Magnesium | SIRS |
| Hemoglobin | White blood cells count | Platelets count |
| Partial Thromboplastin Time | PH - Acidity | PaO2 |
| PaCO2 | Base excess | Bicarbonate |
| Lactate | PaO2/FiO2 Ratio | Oxygen saturation |
| BUN | Creatinine | SGOT |
| SGPT | Total bilirubin | International normalized ratio |
| Prothrombin time | | |

Discrete features

| Age | Gender | Weight |
|---|---|---|
| Mechanical ventilation | | |

SOFA score $\geq$ 2). To extract, preprocess and impute the data we utilize the pipeline described in Komorowski et al. (2018), a process which resulted in 18,585 different patients that comprised our cohort. The cohort is split into three chunks of sizes 70%, 15%, 15% representing the train, validation, and test splits. The observed mortality of the entire cohort was slightly above 6%, and the splits were selected to approximately preserve this ratio. The course of a patient treatment is represented as a trajectory consisting of state-action pairs, terminating upon patient discharge or death. The average trajectory length was 13, with 2 being the smallest and 20 being the largest. The state is a 44 dimensional vector, comprised of 40 time-varying continuous and 4 demographic discrete features, shown in Table 1. Patients' data were discretized using 4h time steps, where variables with multiple measurements within this window were averaged (e.g., heart rate) or summed (e.g., urine output) as appropriate.

## B.3 ACTION SPACE

**Argumentative Policies.** The evidence (action) space of the argumentative policies is defined by the total number of state features, 44 in our case, as listed in Table 1. To prevent the agent from repeating already proposed arguments, we additionally employ *action masking*, setting the log probability of already presented arguments to negative infinity.

**Baseline and Justifiable Policy.** To devise an action space of policies treating sepsis, we follow previous work (Komorowski et al., 2018; Raghu et al., 2017) and focus on managing the total volume of intravenous fluids (IV) and maximum dose of vasopressors (VC) administered to the patient over a 4h discretization window. The dose of each treatment is represented as one of four possible non-null choices derived from observed doses divided into four quartiles. We additionally define another choice, designated as an option 'no drug given'. The combination of these produced 25 possible discrete actions, 5 per each treatment, comprising the action space of the policy.

### B.4 REWARDS

**Sepsis.** To define intermediate and terminal rewards for treating septic patients, following the work of Komorowski et al. (2018), we defined the primary outcome of the treatment via 90-day patient mortality. Therefore, the agent's objective is to optimize for patient survival. To this end, we issue terminal environment rewards of $\pm 15$ for every patient discharge and death, respectively. To stabilize the training of a deep-rl policy, we also issue intermediate rewards that are clinically guided, as in Raghu et al. (2017). These rewards are comprised of fairly reliable indicators of the patient's overall health, namely the SOFA score (measure of organ failure) and a patient's lactate levels (measure of cell-hypoxia, which is usually higher in septic patients). The rewards for intermediate time steps are then shaped as follows:

$$r(s_t, s_{t+1}) = C_0 \mathbb{1}(s_{t+1}^{\text{SOFA}} = s_t^{\text{SOFA}} \& s_{t+1}^{\text{SOFA}} > 0) + C_2(s_{t+1}^{\text{SOFA}} - s_t^{\text{SOFA}}) + C_2 \tanh(s_{t+1}^{\text{Lactate}} - s_t^{\text{Lactate}})$$

where $C_0$, $C_1$ and $C_2$ are tunable parameters which we set to $C_0 = -0.025$, $C_1 = -0.125$ and $C_2 = -2$, following previous work of Raghu et al. (2017). Rewards defined in this way penalize both, the high SOFA scores and lactate values, as well as increases in these quantities.

## C MODELS

### C.1 JUDGE

We defined a judge as a scalar reward function $\mathcal{J}_\theta(a, \{e\}) \in \mathbb{R}$ that quantifies the level of support a decision $a$ has by the set of evidences $\{e\}$. The judge is parametrized by weights $\theta \in \mathbf{R}^{d_1}$ of a neural network with two hidden layers of size 256, using parametric relu (He et al., 2015) activation and batch normalization (Ioffe & Szegedy, 2015). The network takes as input values of proposed evidence $\{e\}$, a binary mask wherein all elements corresponding to the evidence $\{e\}$ are assigned a value of one, while the remaining elements are set to zero, as well as a one-hot encoded action. The addition of a binary mask empirically led to a more stable learning. During training, we augment a tuple $(s_t, a_0, a_1, p) \sim \mathcal{D}$ with an evidence set of state features sampled uniform-random from the state $s_t$ i.e., $\{e\} \sim \mathcal{U}$, anew for each training batch. To learn the parameter $\theta$, we minimize the cross-entropy loss between preference-predictions and labels from the dataset:

$$\min_{\theta \in \mathbb{R}^d} -\mathbb{E}_{(s_t, \{e\}, a_0, a_1, p) \sim \mathcal{D}} \left[ p \cdot \log \mathcal{P}(a_1 \succ a_0, \{e\}) + (1 - p) \cdot \log \mathcal{P}(a_0 \succ a_1, \{e\}) \right]$$

The learning is done for a total of 100 epochs using a batch size of 64, Adam optimizer and a learning rate of 5e-4.

### C.2 ARGUMENTATION

All argumentative models utilize the same network architecture comprised of 2 hidden layers of size 512 with a leaky-relu activation function with slope of 1e-2. The network input consists of a 44 dimensional patient state vector, the decision for which the agent is arguing, as well as a binary mask of arguments proposed thus far. The action space of the agent is represented by all 44 state features (see App. B.3 for more details). For training, we use PPO (Schulman et al., 2017), running the procedure for 1M steps using the Adam optimizer with a learning rate 5e-4 and a batch size of 128. The discount factor was empirically tuned and set to 0.9. The full list of hyperparameters and their considered tuning ranges is given in Table 2. During training, the agent's policy is stochastic: the evidence is sampled from a categorical distribution defined by its logits. To obtain a deterministic policy used in evaluations, we perform an *argmax* operator over obtained logits in a particular state.

#### C.2.1 ISOLATED AGENT

When investigating robustness of the multi-agent debate, we examine two different setups involving an isolated agent baseline: *precommit* (reported in the main paper, Plot 4b) and *adaptive* (reported in the App. D.3, Plot 6b). The former represents an easier case for the agent, but lacks the debate structure. The latter uses a full debate setup as described in Sec. 3.2, but evaluates the isolated agent in a setup slightly different from one it was trained in.

**Precommit.** In this case, an isolated agent is trained to propose evidence $\{e\}$ of size $L/2$ that for a given $a_p$ maximizes $\mathcal{J}'(a_p, \{e\})$, where $\mathcal{J}'$ is a new judge trained to evaluate $L/2$ evidence. When

Table 2: Hyperparameters used for argumentative agents. Unless otherwise indicated, all agents utilize the same parameters.

Common parameters

| Parameter name | Parameter value | Tuning range |
|:---:|:---:|:---:|
| Hidden dim | 512 | [256, 512] |
| Learning rate | 5e-4 | loguniform[1e-5:1] |
| Entropy coefficient | 1e-2 | loguniform[0.00000001:0.1] |
| Clip range | 0.1 | [0.1, 0.2, 0.3, 0.4] |
| Discount | 0.9 | [0.8, 0.9, 0.95, 0.99] |
| GAE lambda | 0.7 | [0.7, 0.8, 0.9, 0.92, 0.95, 0.98, 0.99, 1.0] |
| VF weight | 0.5 | [0.3, 0.5, 0.65, 0.75] |
| Max grad norm | 0.1 | [0.3, 0.5, 0.6, 0.7, 0.8, 0.9, 1, 2, 5] |
| Normalize rewards | true | [true, false] |
| Ortho init | true | [true, false] |

Confuser parameters

| Parameter name | Parameter value | Tuning range |
|:---:|:---:|:---:|
| Hidden dim | 256 | [256, 512] |
| Entropy coefficient | 3e-4 | loguniform[0.00000001:0.1] |
| Clip range | 0.4 | [0.1, 0.2, 0.3, 0.4] |
| VF weight | 0.65 | [0.3, 0.5, 0.65, 0.75] |
| Max grad norm | 2 | [0.3, 0.5, 0.6, 0.7, 0.8, 0.9, 1, 2, 5] |

evaluating robustness (Plot 4b), the agent first proposes $L/2$ evidence, followed by a confuser agent which proposes the remainder.

**Adaptive.** In this case, an isolated agent is trained to propose all $L$ evidence $\{e\}$ that maximize $\mathcal{J}(a_p, \{e\})$, for a given $a_p$. When evaluating robustness (Plot 6b), the isolated agent and its associated confuser take turns and propose a total of $L/2$ evidence each.

### C.2.2 DEBATE AGENTS

For multi-agent scenarios, both *maxmin* and *self-play* agents use the architecture described in the beginning of Sec. C.2, but each modify the underlying optimization pipeline. To train a *self-play* debate agent, we let the agent argue with a (frozen) copy of itself, updating it every 100k steps. This procedure is then repeated for a fixed number of 500 generations. To train the *maxmin* debate agent, we parametrize the agent's opponent with a different set of weights $\phi_2 \in \mathbb{R}^{d_2}$. The procedure starts by training the main agent for 4k steps, followed by training of its opponent for 100k steps. Like in the previous case, the procedure is repeated for 500 generations. This approach is based on the bi-level optimization and allows for overfitting the opponent to the current version of the main agent, which ensures learning of defense strategies against a very strong adversary.

### C.2.3 CONFUSER AGENTS

The evaluation of robustness we presented in Section 5.4 required learning three separate adversaries (one for each argumentative agent), explicitly tasked in providing counterarguments that will confuse the judge. The architecture of these *confuser* agents is mostly the same as that of the argumentative agents, shown in Table 2. For a sample $(s_t, a_0, a_1, p) \sim \mathcal{D}$, the confuser agent is rewarded positively, whenever the judge is convinced of the alternative (non-preferred) action.

Table 3: Hyperparameters used for baseline and justifiable policies. Unless otherwise indicated, all policies use the same parameters. If a parameter does not specify a tuning range, its value has either been selected based on previous work (e.g., buffer $\alpha$ and $\beta$) or tuning was not necessary (e.g., debate coefficient $\lambda$).

Baseline policy

| Parameter name | Parameter value | Tuning Range |
|:---:|:---:|:---:|
| Hidden dim | 128 | [64, 128, 256] |
| Learning rate | 1e-4 | loguniform[1e-5:1e-1] |
| Batch size | 256 | [128, 256, 512, 1024] |
| Polyak update | 1e-3 | [1e-1, 1e-2, 1e-3, 1e-4] |
| ReLu slope | 1e-2 | [1e-1, 1e-2, 1e-3] |
| Discount | 0.99 | - |
| Num. estimation step | 6 | [1, 3, 6, 10, 15] |
| Terminal reward | $\pm 15$ | - |
| Debate scaling coefficient $\alpha$ | $\pm 5$ | - |
| Debate coefficient $\lambda$ | 0.0 | - |
| Buffer $\alpha$ | 0.6 | - |
| Buffer $\beta$ | 0.9 | - |

$\lambda \in [0.25, 0.5, 0.75]$

| Num. estimation step | 3 | - |
|:---:|:---:|:---:|

$\lambda = 1.0$

| Num. estimation step | 1 | - |
|:---:|:---:|:---:|

## C.3 BASELINE AND JUSTIFIABLE AGENTS

Our approach for learning treatment policies for septic patients is based on continuous state-space models and builds on Raghu et al. (2017). The policy is based on a variant of Deep-Q networks (Mnih et al., 2015), specifically double-deep (Van Hasselt et al., 2016) Q-network, also employing the dueling architecture (Wang et al., 2016), where the estimated action-value function for a pair $(s, a)$ is split into separate *value* and *advantage* streams. The final network consists of 2 fully-connected layers of size 128 using leaky-relu activation functions with slope 1e-2. The learning is done in batches of 256 $(s, a, r, s')$ tuples sampled from a Prioritized experience replay buffer (Schaul et al., 2015) with a learning rate of 1e-4. Instead of periodically updating the target network, we leverage Polyak averaging with an update coefficient $\tau$ set to 1e-3. The full list of used hyperparameters is given in Table 3. Similar to Raghu et al. (2017), we augment the standard Q-network loss. First, we added a regularization term that penalizes Q-values outside the allowed threshold $Q_{\text{thresh}} = \pm 20$. In addition, we clip the target network outputs to $\pm 20$, which empirically proved to stabilize the learning. The final loss function we used is given by:

$$\mathcal{L}(\Theta) = \mathbb{E}\left[(Q_{\text{double-target}} - Q(s, a; \Theta))^2\right] + \beta \cdot \max(|Q(s, a; \Theta)| - Q_{\text{thresh}}, 0)$$
$$Q_{\text{double-target}} = r + \gamma \cdot Q(s', \arg\max_{a'} Q(s', a'; \Theta); \Theta')$$

Where $\beta$ is a user-specified coefficient we set to $\beta = 5.0$ in all our experiments following Raghu et al. (2017). The learning is done for a total of 25k iterations, evaluating the policy every 50 iterations using weighted importance sampling (WIS) (Sutton & Barto, 2018) on a held-out test set.

**Behavioral Policy.** The calculation of the importance sampling ratio requires access to a so-called *behavioral policy* that generated offline samples. To obtain it, we train a behavior-cloning (BC) clinician policy to take in a state $s_t$ from the patient cohort and predict the action $a_t$ taken by the human clinician, minimizing the cross-entropy loss over the training dataset. The network consists of two fully-connected layers of size 64. We run the training with the Adam optimizer, using a learning rate of 1e-3 and a weight decay set to 1e-1 for a total of 100 epochs with a batch size of 64.

# D ADDITIONAL RESULTS

## D.1 SHORTER DEBATES

In the main text, we have seen positive results involving debate using 6 arguments, which amounts to $\sim 13\%$ of the entire state. In this section, we want to further examine the effectiveness of debate when limiting the number of evidence to $L = 4$, amounting to 9% of the entire state.

The first question that arises when further limiting the amount of state visibility during debate is the impact it has on the quality of learned task policies. In Plots 5a-5d, we evaluate the performance of different justifiable policies on a held-out test set during the course of training using weighted importance sampling, for the case of 4 and 6 evidence limit. While the reduced number of evidence exposes the judge to only 9% of the entire state, we can see that the achieved performance is comparable to the case where $L = 6$. Furthermore, in Plot 5f, we also confirm that trained policies are significantly more preferred to the baseline policy, a trend equivalent to the one we saw in the main text. Apart from these quantitative evaluations, in Plots 5g and 5h, we show the qualitative analysis of patient mortality from Sec. 5.2. We confirm that the lowest mortality is observed when clinician prescribed doses recommended by justifiable policies, thus further signaling potential validity of policies trained with rewards stemming from debates with $L = 4$ evidence.

To evaluate the effectiveness of argumentative agents, we repeat the evaluation from Sec. 5.4. When exposed to 4 randomly selected evidence, the judge trained via the procedure outlined in App. C.1 achieves accuracy of $\sim 59\%$. Plot 5e shows the judge's accuracy when different argumentative agents propose $L = 4$ evidence [12], averaged over 1000 samples $(s_t, a_1, a_2, p) \sim \mathcal{D}$. Without the confuser, all three agents achieve performance similar to the case of $L = 6$ evidence, boosting the judge's accuracy to almost 90%. In the setup involving an adversary, agents propose a total of 2 evidence each before the judge evaluates the outcome. The observed trend is also similar to the one from main text (Sec. 5.4).

## D.2 PREFERENCE BREAKDOWN

To gain a better understanding about the behavior of learned justifiable policies, we further examine actions they propose. In particular, Plot 6a shows the percent of times an action from the justifiable policy was preferred to that of the baseline policy (JP), percent of times when the action of the baseline policy was preferred to the action of the justifiable policy (BP), and percent of time when the two were equally preferred (EP) [13]. We see that the number of actions proposed by the justifiable agent and the baseline agent that are equally preferred decreases as the parameter $\lambda$ is increased: for $\lambda = 0.25$ two agents chose the equally justifiable action about 49% of the times, for $\lambda = 0.50$ this number drops to 40%, whereas for $\lambda = 0.75$ and $\lambda = 1.0$ this number is 36% and 35% respectively. Out of the remaining actions, the ones from justifiable policies were increasingly more preferred, as the parameter $\lambda$ was increased.

## D.3 ISOLATED AGENT SETUP

In App. C.2.1, we described two setups which we consider when evaluating robustness of the isolated agent, namely *precommit* and *adaptive*. In Plot 6b, we show the accuracy of the judge in predicting the preferred action when evidence was proposed by an isolated agent trained with one of these approaches for 1000 different debate games. Without the confuser, both approaches amplify the capabilities of the judge, performing roughly equivalent. When faced with a confuser agent, the precommit approach performs better, since in the adaptive case, the agent is evaluated in a debate-like setup, which was not accounted for during training. In both cases, however, we observe that the isolated agent is not robust to an adversary.

## D.4 ALTERNATIVE UTILITY FUNCTIONS

In Sec. 3.2, we have defined the utility function $\mathbb{U}$ to output binary values $\{-1, 0, +1\}$ based on the justifiability rewards obtained from the judge. One might ask if there are alternative ways of

---

[12]Like in the main text, we use the *precommit* setup for the isolated agent (see App. C.2.1 for more details).

[13]In our experiments, the judge deemed two actions equally justifiable whenever they were the same.

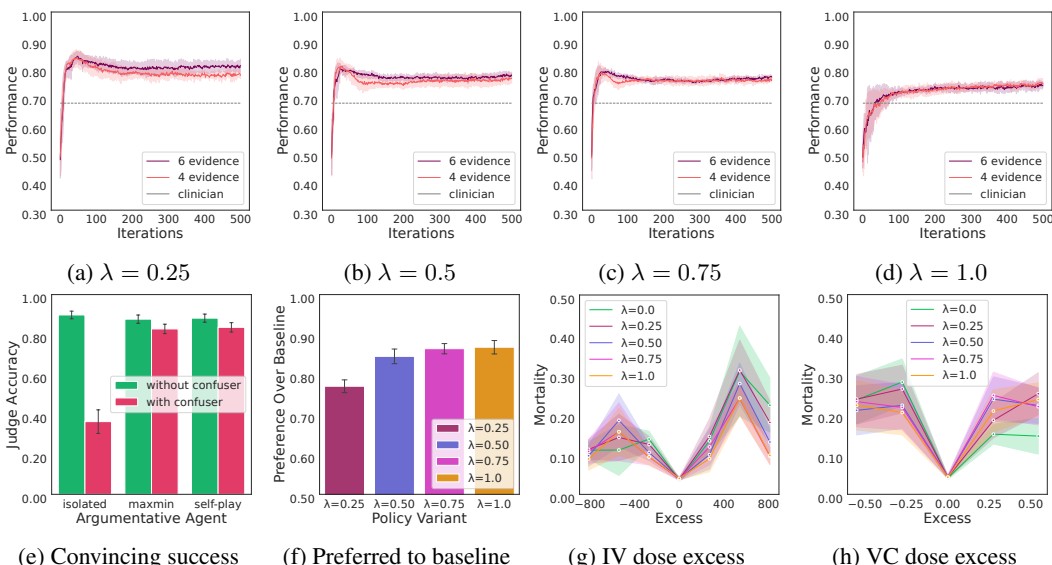

(a) $\lambda = 0.25$     (b) $\lambda = 0.5$     (c) $\lambda = 0.75$     (d) $\lambda = 1.0$

(e) Convincing success    (f) Preferred to baseline    (g) IV dose excess    (h) VC dose excess

Figure 5: Quantitative and qualitative evaluation of policies trained using debate-based rewards limited to $L = 4$ evidence. For (f)-(h), the confidence intervals represent $\pm 2$ standard errors of the mean over 5 random seeds. (a)-(d) Performance of policies trained with $L = 4$ and $L = 6$ evidence, as measured by WIS evaluation on a held-out test set with $\pm 1$ terminal rewards for every patient discharge or death. The mean and standard deviation are reported over 5 random seeds. (e) Accuracy of the judge in predicting the preferred action using 4 proposed evidence, with and without the confuser agent. The CI represent $\pm 2$ standard errors of the mean estimate. (f) Percent of times judge preferred decisions of justifiable policies (i.e., $\lambda > 0.0$) compared to those of the baseline policy (i.e., $\lambda = 0.0$). (g) (h) Observed patient mortality (y-axis) against variations in IV/VC treatment doses prescribed by clinicians compared to the recommendations of learned policies (x-axis).

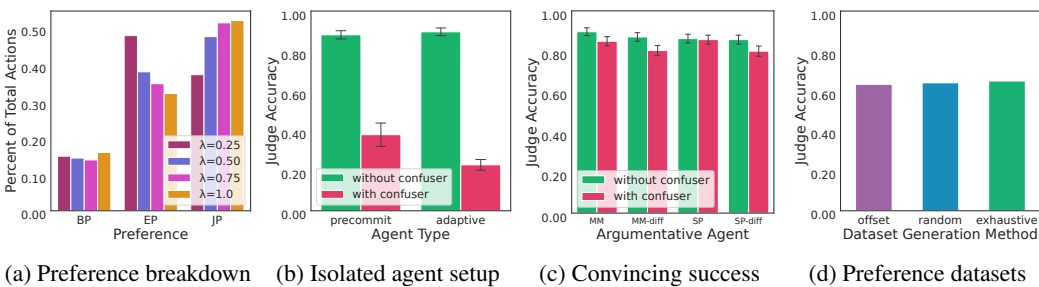

(a) Preference breakdown    (b) Isolated agent setup    (c) Convincing success    (d) Preference datasets

Figure 6: (a) Percent of times when actions of the justifiable policy were more preferred to those of the baseline policy (JP), less preferred (BP) and equally preferred (EP). (b) Accuracy of the judge in predicting the preferred action, with and without the confuser agent when evidence was proposed by two isolated agents trained in *precommit* and *adaptive* setup. The CI represents $\pm 2$ standard errors of the mean estimate. (c) Accuracy of the judge in predicting the preferred action, when evidence was proposed by maxmin (MM) and self-play (SP) agents trained using the utility function proposed in Sec. 3.2 compared to agents trained using the utility function defined as a difference in predicted rewards proposed in App D.4, MM-diff and SP-diff respectively. The CI represents $\pm 2$ standard errors of the mean estimate. (d) Accuracy of the judge in predicting the preferred action with evidence sampled uniform-random, trained on datasets generated with different methods.

defining $\mathbb{U}$ that preserve the zero-sum structure and potentially positively influence the learning. In this section, we consider one particular alternative that defines the utility function based on the difference between predicted rewards, which might provide a more informative learning signal. In particular, we set $u_1(n_L) = -u_2(n_L) = \mathbb{U}(a_t, a_t^B, \{e_{n_L}\})$, with $\mathbb{U}$ defined as $\mathbb{U}(a_t, a_t^B, \{e\}) = \mathcal{J}(a_t, \{e\}) - \mathcal{J}(a_t^B, \{e\})$. We rerun the experiments from Sec. 5.4 for maxmin and self-play agents and show preference recovery success and robustness results in Plot 6c. We can see that the two approaches perform similarly in situations which do not involve a confuser agent. However, it seems that defining the utility using the difference in judge's rewards leads to slightly lower scores when debate agents are faced with an adversary.

## D.5 ALTERNATIVE PREFERENCE DATASET

In Sec. 5.1 we construct a preference dataset by matching every pair $(s_t, a_t)$ from the cohort with an alternative action $a_r \sim \mathcal{U}(A), a_r \neq a_t$ sampled uniform-random from the action space. In this section, we examine additional strategies one could take when constructing such a synthetic dataset. In particular, we consider the *random* strategy we just described, paired with two alternative ones, namely *exhaustive* and *offset*. The exhaustive strategy pairs $a_t$ with all possible alternative actions, 24 in total. The offset strategy pairs $a_t$ with an alternative action that is in its neighborhood. To define a neighborhood, we recall that there are a total of 5 choices for both, vasopressors (VC) and intravenous fluids (VC). Therefore, we can write $a_t = 5 * \text{IV} + \text{VC}$, where IV, VC $\in \{0, 1, 2, 3, 4\}$. To obtain an alternative action, we consider changing IV and VC by an offset sampled uniform-random from a set $\{-1, 0, 1\}$, for both IV and VC. We then train a new judge for each of the datasets and show its accuracy in Plot 6d. The exhaustive variant represents the most informative, but also unrealistically large, dataset. The random variant represents somewhat of a "middle ground" in terms of dataset difficulty. Lastly, the offset variant represents the most difficult case, as differences between two actions are more nuanced. However, while the achieved accuracies reflect the difficulty of the corresponding dataset, the capabilities of a trained judge model are roughly the same.

