# OpenReview forum: "Reward Design for Justifiable Sequential Decision-Making"
_ICLR.cc/2024/Conference — ICLR 2024 poster_

### Official Review · Reviewer_pkzb · 2023-10-17

**Soundness:** 3 good
**Presentation:** 4 excellent
**Contribution:** 3 good
**Rating:** 6
**Confidence:** 2

**Summary:**

The paper introduces a date framework for reward-shaping and quantifying the justifiability of a decision at a given state. An application is outlined for treating sepsis patients.

**Strengths:**

* Paper is clear and straight to the point.

* The authors provide adequate motivation for their model, which is quite natural.

* I believe that the problem the authors attempt to address is important for any practical use of RL.

**Weaknesses:**

* No real weakness. Some slight inaccuracies in the terminology/in some definitions (see my questions below) but this should be easily fixable.

**Questions:**

1. Bottom of page 1, “the human as a suboptimal decision maker”: do you mean that the RL agent is optimal? But if optimality refers to the actions deployed in practice, the RL agent will not be optimal either, no?

2. The authors in [A,B] also consider sequential decision-making (MDPs)  influenced by a baseline policy and an agent to model partial adherence to an algorithmic recommendation. Is the setup in this submission close to the setup in [A,B]?

3. Definition of $R_{J}$ : what is the expectation over? The transition probabilities (since policies are deterministic)? Is the decision to be optimized $a_{t}$ or $a^{B}_{t}$ ?

4. p4: the definition of $\mathbb{U}$ introduces the notation $\mathcal{J}$ but the next sentence uses $J$. Please clarify.

5. Section 4.4: can you please justify the exact architecture and neural nets used in the paper (two-fully connected layer, size 256, choice of parametric relu, etc.)? This section appears very ad-hoc, which it may be, and that is fine, but it requires some discussion/empirical justification.

[A] Grand-Clément, J., & Pauphilet, J. (2022). The best decisions are not the best advice: Making adherence-aware recommendations. arXiv preprint arXiv:2209.01874.

[B] Faros, I., Dave, A., & Malikopoulos, A. A. (2023). A Q-learning Approach for Adherence-Aware Recommendations. arXiv preprint arXiv:2309.06519.

---

> ### Author Response · Authors · 2023-11-16
> **Response to Reviewer pkzb**
>
> Thank you for your comments and questions. We are glad to see you consider the problem of justifiable decision-making to be of high practical importance, and that you consider our presentation to be clear and straight to the point. We are also thankful for the references you have provided, and are happy to discuss them further. In the following, we try to answer all of your questions and comments. The key points of our response can be summarized as follows:
> - We discuss similarities and differences to the two works you kindly suggested;
> - We answer your questions about optimality of deployed policies, definition of $R_J$ and choices of model architectures.
>
> ## Discussion on General Comments
>
> > Bottom of page 1, “the human as a suboptimal decision maker”: do you mean that the RL agent is optimal? But if optimality refers to the actions deployed in practice, the RL agent will not be optimal either, no?
>
> What we intended to say there is that a human may not act optimally or rationally in every situation, as it is inherently limited in capacity. To extend on this, our line of reasoning touched upon the fact that in certain situations, a human may not be able to comprehend the full state available to the acting agent, which may hinder its judgment. We also acknowledge your comment about optimality of the deployed agent. Indeed, the agent is learned, and will therefore also be suboptimal, although the referenced sentence does not imply any assumptions about this. Lastly, we have slightly expanded Sec. 3.1 to better clarify the assumptions made by our framework about optimality and choice of the baseline policy.
>
> > The authors in [A,B] also consider sequential decision-making (MDPs) influenced by a baseline policy and an agent to model partial adherence to an algorithmic recommendation. Is the setup in this submission close to the setup in [A,B]?
>
> Thank you very much for sharing these works. We have also updated the related work section to include them. Regarding [A], we find a conceptual similarity in that both approaches aim to improve what is assumed to be a given baseline policy. However, [A] operates in a context of expert-in-loop setting, where a baseline policy is considered to represent a human decision-maker and a policy to be learned (in their terms, recommendation policy) aims to adapt its actions to it. Another prominent difference is found in a way the environment evolves: in [A] both the baseline policy and the learned policy can influence dynamics of the environment. Differently, in our setup, baseline policy never influences state transitions: the justifiable agent is a sole actor in the environment. Likewise, authors in [B] examine a setting similar to the one in [A], which also exhibits mentioned similarities and differences. As a general note, both [A] and [B] consider a different problem setup, where the primary goal is to improve and adapt the AI policy to the baseline policy representing a human, which is the main decision-maker in the environment. In contrast, we consider a problem of improving the baseline policy in a setup where a learned (i.e., justifiable) policy is the main actor in the environment, and a baseline policy is only used as a reference point to improve upon.
>
> > Definition of $R^J$: what is the expectation over? The transition probabilities (since policies are deterministic)? Is the decision to be optimized $a_t$ or $a_t^B$?
>
> In the definition of the justifiability reward $R^J$, the expectation is over the trajectories, which are stochastic due to non-deterministic transitions. The decision to be optimized is the one coming from the justifiable policy, namely $a_t$, whereas the decision $a_t^B$ is coming from a (fixed) baseline policy. We have also slightly updated the paper to make this part more easily understandable to the reader.
>
> > p4: the definition of U introduces the notation J but the next sentence uses J. Please clarify.
>
> Thank you for noticing inconsistencies in our presentation here. The definition of U introduces J as a model of a human judge. Later mentions also refer to the same entity. We have corrected the paper accordingly.

---

> > ### Author Response · Authors · 2023-11-16
> > **Response to Reviewer pkzb (cont.)**
> >
> > > Section 4.4: can you please justify the exact architecture and neural nets used in the paper (two-fully connected layer, size 256, choice of parametric relu, etc.)? This section appears very ad-hoc, which it may be, and that is fine, but it requires some discussion/empirical justification.
> >
> > Thank you for raising this concern. The architecture and hyperparameters used for our argumentative agents and the judge model were obtained through careful hyperparameter tuning procedures, as we had no prior work to rely on. We did, however, include a full list of hyperparameter tuning ranges and further details about the model’s architecture (e.g., insights about including a binary mask as part of the judge’s input, usage of action masking, etc.) as part of our appendix. For policies treating sepsis, we designed our models on the basis of [1], and we have now also included a list of their hyperparameter tuning ranges in App. C.3.
> >
> > ## Conclusion
> > We thank you again for your comments and questions. We are happy to answer anything else in addition.
> >
> > ## References
> > 1. A. Raghu, M. Komorowski, I. Ahmed, L. Celi, P. Szolovits, and M. Ghassemi, “Deep Reinforcement Learning for Sepsis Treatment.” arXiv:1711.09602

---

> > > ### Author Response · Authors · 2023-11-21
> > > **Closing of the Discussion Phase**
> > >
> > > Thank you once again for your feedback. As the rebuttal/discussion phase ends soon, we wanted to check if you had any additional comments. We hope that our response and the updates in the paper address your concerns. We would be happy to answer any further questions.

---

### Official Review · Reviewer_2TSd · 2023-10-30

**Soundness:** 3 good
**Presentation:** 3 good
**Contribution:** 3 good
**Rating:** 6
**Confidence:** 3

**Summary:**

General Approach:
This paper introduces a method of learning policies for sequential decisions that are justifiable to a (human) judge. The general technique is to use reward shaping, where the reward of an RL agent is determined jointly by the environment and by the judge, and parameterized by a coefficient that determines the balance between performance (determined by environment) and justifiability (determined by the judge). The novel aspect of the approach is the construction of the reward from the judge using a debate game. The reasons given for using a zero-sum debate game in extensive form is to ensure that (1) the evidence presented by the agent is sufficient yet concise, and (2) the argument (collection of evidence) should not be easy to refute by providing additional evidence to the judge.

Debate Game:
The debate game is a zero-sum, perfect-information game between a baseline agent and an agent trying to learn more justifiable decisions, where the strategy space is composed of the evidence that each agent can introduce at each stage in the game, and the utilities/payoffs are determined by the judgement of the judge. Importantly, the number of steps in the game, and hence the size of the ultimate evidence set, is much smaller than the set of possible elements that can be added as evidence. The judge provides real-valued judgements J(action, evidence set). Two two agents each suggest an action, and then take turns introducing elements to evidence to justify their action, so they end up each introducing half of the evidence in the evidence set. The judge then judges each action with respect to the evidence set, and the agent whose action is preferred by the judge based on the evidence gets a utility of +1 (and the other -1). The reward from the judge is then proportional to the value of the game (or, rather, an approximation of it).

An empirical study is done on data for patients with Sepsis, including comparison to trainign with only environment-based and only debate-based rewards and agents who learn without an adversary.

**Strengths:**

Overall, the paper is well-organized, the problem is well-motivated, and some of the empirical results are highly compelling (i.e. Figs 2c and 2d). This approach provides one way of capturing the tradeoff between learning an optimally performing policy and one that is justifiable to a human agent, creating the ability to interpolate in between. The use of a debate game is a novel contribution the should, in theory, help train the RL agent to produce decisions whose justifications achieve robustness in the face of new information.

**Weaknesses:**

There are some weaknesses in the methodology and presentation of the work, but these are relatively minor. My biggest concern are the ethical considerations that are not addressed. My review would be to accept if not for the severe ethical implications of the work that have gone unstated.

Methodology and Presentation:
The model of the game seems more complex than necessary, and bakes in assumptions that don’t have clear motivation. Specifically, both agents must provide equal amounts of information, in alternating fashion. Why not simply have a game where the justifying agent provides some number of pieces of evidence L and the adversarial (baseline or confuser) agent provides any number of additional pieces of information? What is the benefit of creating a game with L stages where the agents alternate with best response? The paper motivates the use of Nash Equilibria to ensure robustness against additional information, but no motivation is given for this multi-stage debate game instead of letting each agent make 1 move each.

A missing benchmark is how often the benchmark and justifying agents choose the same action. This would be highly informative.


Unless I misunderstand the methodology, the benchmark with the isolated agent does not seem entirely appropriate. The isolated agent is trained to provide L pieces of evidence and then tested in a setting where it provides L/2 and a confuser provides L/2. It is not surprising that it is vulnerable to a confuser that provides the evidence least-supportive of its L/2 pieces of information. A better benchmark would be to train the isolated in a setting where it only provides L/2 pieces of information, and then see how robust it is to the setting where L/2 adversarially selected pieces of information are added.

In Fig 2a and 3a-d, the clinician’s performance is compared to the learned policies according to WIS. But why would WIS be a reasonable way to measure the performance of the real-world clinician based on clinical decisions?

The payoff function G is introduced, but it is never specified how it relates to the utility function U.

The use of the “Bradley-Terry model” (based on the Boltzmann distribution) is never given a justification.

Also, in the real world, not all pieces of evidence have equal complexity to evaluate, not all combinations of evidence have equal complexity to evaluate.

Minor Errors:

Typo: (Section 3.1) \gamma^{i} instead of \gamma^t in multiple places

Typo (Section 3.2):  after the definition of U the text uses J instead of \mathcal{J}

There doesn’t seem to be any need to introduce the notation \mathcal{E}(n). Just use \mathcal{E} \backslash n.

Typo (Section 5.3, p2): “…even tough” —> “…even though”

Typo (Figure 4): The legend labels for “with confused” and “without confused” appear to be inconsistent with the plots.

**Questions:**

Isn’t the value of the game always in {1, -1}?

Why does J need to be real-valued? Isn’t it enough just to be a weak order over action-evidence pairs? The actual value of J seems irrelevant to the model.

**Details Of Ethics Concerns:**

The method being designed is proposed for a beneficial task - improving medical treatment. However, it seems as though it would be just as easily used (and perhaps more effective) to implement it for nefarious purposes. For example, consider a chatbot being trained to get victims to reveal private personal information. The chatbot seeks to learn a strategy that maximizes value from the information victims reveal and learns what information/evidence/text to provide the agents that seems most compelling to them. It provides the arguments that are most difficult for the victims to refute, and learns from the arguments others have provided as responses in the past. In that case the judge and the confuser are the same agent. The problem is that the “performance” may be based on a nefarious objective, and ultimately the debate reward simply trains the justifying agent how to most effectively deceive the judge. One can imagine many more similarly nefarious scenarios that come from this reward-shaping approach, and learning to maximize justifiability in general. Without constraints, or formal statements about what information must be provided to ensure transparency, learning justifiability necessarily implies learning deception. Static methods like SHAP that do not try to learn justifiability do not pose a similar risk.

---

> ### Author Response · Authors · 2023-11-16
> **Response to Reviewer 2TSd**
>
> Thank you for your detailed review, perceptive comments and questions. We are glad to see you find the problem to be of importance and that you think our method has merits. In the following, we try to answer all of your questions and comments. The key points of our response can be summarized as follows:
> - We address your comment about ethics concerns pertaining to our method;
> - We address your questions and concerns about an additional benchmark comparing actions proposed by the justifiable and baseline agents, and confuser evaluation of the isolated agent with additional experiments;
> - We address your questions about the complexity of the debate and evaluation of evidence, payoff function G, evaluation of the clinician and choice of the Bradley-Terry model.
>
> ## Questions Involving Additional Experiments
>
> > A missing benchmark is how often the benchmark and justifying agents choose the same action. This would be highly informative.
>
> Thank you for pointing this out. We acknowledge that there is an opportunity to provide richer information to the reader in this regard. To address this, in App. D.2 and Plot 6a, we show the percent of times an action from the justifiable policy was preferred to that of the baseline policy (JP), percent of times an action of the baseline policy was preferred to that of the justifiable policy (BP) and percent of times when the two were equally preferred (EP). In addition, we updated the Plot 2b to show the percent of times an action from the justifiable policy was preferred to that of the baseline policy, but only when the two differ. In our experiments, the judge deemed two actions equally justifiable only when they were the same.
>
> > Unless I misunderstand the methodology, the benchmark with the isolated agent does not seem entirely appropriate. The isolated agent is trained to provide L pieces of evidence and then tested in a setting where it provides L/2 and a confuser provides L/2. It is not surprising that it is vulnerable to a confuser that provides the evidence least-supportive of its L/2 pieces of information. A better benchmark would be to train the isolated in a setting where it only provides L/2 pieces of information, and then see how robust it is to the setting where L/2 adversarially selected pieces of information are added.
>
> Thank you for raising this concern. The isolated argumentative agent was initially trained to propose L=6 evidence, whereas during evaluation against a confuser, it was proposing L=3 evidence in a debate-like setup. As a response to your comment, we introduce a new setup for the isolated agent (also described as part of the changes made to the paper in our general response). In the new version, the isolated agent is trained to propose L=3 evidence. When evaluating robustness, we do not perform debate, but rather let the agent first propose L=3 evidence, followed by a confuser proposing the remainder. We report the new results in the main text (Sec. 5.3, Plot 4b). We also include the previous setup and do a comparison to the updated one in App. C.2.1 and App. D.3. The conclusions we drew about robustness of the isolated agent in Sec. 5.4 remain the same in both cases.

---

> > ### Author Response · Authors · 2023-11-16
> > **Response to Reviewer 2TSd (cont.)**
> >
> > ## Discussion on General Questions and Paper Weaknesses
> >
> > > The model of the game seems more complex than necessary, and bakes in assumptions that don’t have clear motivation. Specifically, both agents must provide equal amounts of information, in alternating fashion. Why not simply have a game where the justifying agent provides some number of pieces of evidence L and the adversarial (baseline or confuser) agent provides any number of additional pieces of information? What is the benefit of creating a game with L stages where the agents alternate with best response? The paper motivates the use of Nash Equilibria to ensure robustness against additional information, but no motivation is given for this multi-stage debate game instead of letting each agent make 1 move each.
> >
> > The multi-step debate was chosen to best reflect a real-world argumentative procedure and is also based on the prior work [1]. The multi-step debate, or more specifically, a full path in the debate game tree, holds additional information that can be used to judge the outcome. From the perspective of a human, seeing the order of proposed evidence might give additional information about their relationship and mutual level of attack/defense that can be used to issue judgments [6]. Note that we have not fully exploited the structure of the multi-step debate in this work. In particular, the judge model is invariant to different paths that all lead to the same leaf node. Future work could consider approaches that can leverage the additional information provided by the multi-step debate framework. It is also important to point out that a general debate framework, as described in our work and in [1], is meant to address cases where a human judge issues judgments over proposed evidence, instead of the full state. It is therefore paramount to allow both agents to express their for/against evidence until a judge can confidently determine the winner. Having one agent propose more/less evidence would give them an unfair advantage, which could further be exploited by an adversary.
> >
> > > In Fig 2a and 3a-d, the clinician’s performance is compared to the learned policies according to WIS. But why would WIS be a reasonable way to measure the performance of the real-world clinician based on clinical decisions?
> >
> > We would like to emphasize that we do not use WIS to evaluate the performance of a clinician. Instead, the performance of a clinician reported in the mentioned figures is the observed reward from the entire (test) dataset, meant to serve as a guideline, similar to [2]. We describe this when introducing the experimental setup in Sec. 5.1, paragraph “Baselines”. We have also slightly updated this part to better communicate this.
> >
> > > The payoff function G is introduced, but it is never specified how it relates to the utility function U.
> >
> > The payoff function is simply the utility of the first player, conditioned on both players following a given strategy profile. We have slightly updated the text to make this more clear.
> >
> > > The use of the “Bradley-Terry model” (based on the Boltzmann distribution) is never given a justification.
> > The choice of a Bradley-Terry model was inspired by the seminal work that examined specification of the reward function for reinforcement learning agents via preference ratings over trajectories [5]. It is a standard model for estimating score functions from pairwise preferences. In our framework, it can be understood in the context of equating judge’s rewards with a justifiability ranking scale. In particular, the difference in judge’s reward of two actions estimates the probability that a human considers one more justified than the other, given a particular set of evidence.
> >
> > > Also, in the real world, not all pieces of evidence have equal complexity to evaluate, not all combinations of evidence have equal complexity to evaluate.
> >
> > Thank you for the comment. In practice, both individual evidence and their combination might have varying difficulties when evaluated by the human. We hypothesize that training agents via debate forces them to propose the most relevant evidence that is sufficient to render a decision justifiable, but we acknowledge that this does not necessarily imply that the evidence is easily judged by the human. We do, however, note that debate requires the human to evaluate significantly less information than contained in a full state, which in itself eases the judging process and enables scaling to problems where full state comprehension is beyond the scope of a human judge (e.g., debate [1], recursive reward modeling [3] and iterated amplification [4] all address a similar problem). Future research could also incorporate the difficulties of evaluating a particular set of evidence as an additional feedback signal and therefore require agents to provide evidence that is more easily judged, in addition to being the most representative and justifying of the decision.

---

> > > ### Author Response · Authors · 2023-11-16
> > > **Response to Reviewer 2TSd (cont.)**
> > >
> > > > Minor Errors
> > >
> > > Thank you for noticing these errors. We have incorporated your feedback into the revised paper.
> > >
> > > > Isn’t the value of the game always in {1, -1}?
> > >
> > > Thank you for the comment. We apologize for the confusion that may have occurred due to our mistake in writing of the utility function (Sec. 3.2), which we also elaborated in our general response at the top of this page. Throughout the paper, we assumed that when two actions are equally justifiable (e.g., as in a case when they are equal) the debate game draws, ending in both players obtaining a score of 0. Therefore, the value of the game is always in {-1, 0, +1}.
> > >
> > > > Why does J need to be real-valued? Isn’t it enough just to be a weak order over action-evidence pairs? The actual value of J seems irrelevant to the model.
> > >
> > > In our framework, the judge model follows the Bradley-Terry (BT) model, as in [5], which in turn implies it is unbounded and real-valued. The BT model is a standard model for estimating score functions from pairwise preferences, and we adopt it as such in our work. Examining and comparing alternative models that encapsulate a weak order over action-evidence pairs is an interesting direction for future research.
> > >
> > > ## Discussion on Ethical Considerations
> > >
> > > > The method being designed is proposed for a beneficial task - improving medical treatment. However, it seems as though it would be just as easily used (and perhaps more effective) to implement it for nefarious purposes. For example, consider a chatbot being trained to get victims to reveal private personal information. The chatbot seeks to learn a strategy that maximizes value from the information victims reveal and learns what information/evidence/text to provide the agents that seems most compelling to them. It provides the arguments that are most difficult for the victims to refute, and learns from the arguments others have provided as responses in the past. In that case the judge and the confuser are the same agent. The problem is that the “performance” may be based on a nefarious objective, and ultimately the debate reward simply trains the justifying agent how to most effectively deceive the judge. One can imagine many more similarly nefarious scenarios that come from this reward-shaping approach, and learning to maximize justifiability in general. Without constraints, or formal statements about what information must be provided to ensure transparency, learning justifiability necessarily implies learning deception. Static methods like SHAP that do not try to learn justifiability do not pose a similar risk.
> > >
> > > Thank you for your thoughtful consideration of the ethical implications of our work. To address your concerns, we have now included a discussion on ethical considerations in Sec. 6 of the paper. In the following, we also further elaborate on this.
> > >
> > > As discussed in [1], the debate game presumes that it is more challenging to argue for an action that is misaligned with the objective of the human evaluator (judge) than it is to refute this action. Hence, the debate framework provides a recipe for how to defend against malicious actors: train helper argumentative agents that aim at refuting the proposed claims. In our setting, confuser agents (Section 5.4) could act as helper agents. In this case, confuser agents would be refuting a misaligned action, so presumably they would have high success rates. Going back to your example that considers a malicious chatbot, this means that one approach to defending victim agents against deceptive behavior could be to train a helper chatbot whose goal is to refute the arguments of the malicious chatbot. We deem this to be an interesting research direction, but also orthogonal to the aspects we study in this work.
> > >
> > > ## Conclusion
> > > We thank you again for your comments and questions. We are happy to answer anything else in addition.
> > >
> > > ## References
> > > 1. G. Irving, P. Christiano, and D. Amodei, “AI safety via debate.” arXiv:1805.00899
> > > 2. M. Komorowski, L. A. Celi, O. Badawi, A. C. Gordon, and A. A. Faisal, “The Artificial Intelligence Clinician learns optimal treatment strategies for sepsis in intensive care,” Nat Med, vol. 24, no. 11, Art. no. 11, Nov. 2018.
> > > 3. J. Leike, D. Krueger, T. Everitt, M. Martic, V. Maini, and S. Legg, “Scalable agent alignment via reward modeling: a research direction.” arXiv:1811.07871
> > > 4. P. Christiano, B. Shlegeris, and D. Amodei, “Supervising strong learners by amplifying weak experts.” arXiv:1810.08575
> > > 5. P. Christiano, J. Leike, T. B. Brown, M. Martic, S. Legg, and D. Amodei, “Deep reinforcement learning from human preferences.” arXiv:1706.03741
> > > 6. P. M. Dung, “On the acceptability of arguments and its fundamental role in nonmonotonic reasoning, logic programming and n-person games,” Artificial Intelligence, vol. 77, no. 2, pp. 321–357, Sep. 1995.

---

> > > > ### Author Response · Authors · 2023-11-21
> > > > **Closing of the Discussion Phase**
> > > >
> > > > Thank you once again for your feedback. As the rebuttal/discussion phase ends soon, we wanted to check if you had any additional comments. We hope that our response and the updates in the paper address your concerns about the ethical implications of this work. We would be happy to answer any further questions.

---

> > ### Comment · Reviewer_2TSd · 2023-11-22
> > **Additional Experiments / Author Response**
> >
> > The additional work you provided addresses both of my concerns in full. I also found your responses to other reviewers to be generally thoughtful. This will be reflected in my final review.
> >
> > It is good to see an explicit recognition of the theoretical considerations. Of course, the acknowledgment is important, but there remains the ethical question of whether such research directions should be pursued and published given the potential for abuse. I hope the authors take these risks into account when deciding what problems to work on, and not only acknowledge them when publishing.

---

> > > ### Author Response · Authors · 2023-11-23
> > > **Thank you for your comment**
> > >
> > > Thank you for your comment. We are happy to see that the additional related work addresses your concerns, and that you also appreciate our response to other reviewers. We are also thankful for your additional remarks, and we recognize their importance. We initially chose to build on the line of work involving debate, believing that advancing such multi-agent training methods holds the promise of creating effective, safe, and aligned systems. While this does not preclude a potential for misuse, we hope that our approach will further advance the development of systems robust to adversaries and aligned with human interests.

---

### Official Review · Reviewer_8w5y · 2023-11-01

**Soundness:** 2 fair
**Presentation:** 3 good
**Contribution:** 3 good
**Rating:** 8
**Confidence:** 2

**Summary:**

This work describes a framework for producing RL agents which take "justifiable" actions. This is done by defining a "debate game" for a state and a pair of actions, in which two adversarial debaters (each representing one of the actions) take turns picking "evidence" (aka features or factors) from the state, each trying to maximize the chance that a synthetic "judge" which takes in all the evidence as input will rate their action higher. A justifiable action is thus an action for which the outcome of the action's debate game does indeed result in the judge rating the action highly.

The paper describes experiments performed in the RL environment of providing interventions for patients with Sepsis. The work compares a baseline (a standard RL agent trained to maximize reward in the environment) with the novel approach (an RL agent trained to maximize a combination of reward from the environment and reward from the judge's judgment of the result of the debate game for the chosen action vs. the baseline action). The optimal policies for the debate game are trained using self-play RL (the debate game is perfect-information). There is a dataset containing real clinicians' interventions in the Sepsis setting. The judge is trained to predict, for each state in the dataset, the clinician's actual action based on evidence which is sampled uniformly random from the available evidence.

The work is highly related to explainability. Its methodology is similar to research in learning pairwise preferences.

Many experiments are performed, and there is ample discussion about motivations and future research.

---

Post-rebuttal period update: I originally rated this paper a 6. The authors addressed some comments I made, and performed additional experiments. If I could, I would raise my rating to something between 6, and 8 (but unfortunately such an integer does not exist). I still don't like that the experimental design doesn't exactly match the motivation, but the updated version more explicitly acknowledge this, and the paper is still an interesting proof-of-concept. Finally, with the addition of the new experiments, the paper contains so many experiments to investigate questions that one could want to know the answer to, that its value passes the bar for publication. Therefore, I decided to break the tie between 6 and 8 by rating the paper an 8.

**Strengths:**

This is my first time hearing about debate agents, so my to eyes the very idea of using debate to train aligned agents is novel. However, this may not have been true if I had previously read one of the Debate papers in the Related Work section. Still, the research direction seems exciting, and this paper is a solid proof-of-concept exploring the specific idea of using a judge's evaluation of the result of a debate game as RL reward.

There were many experiments performed, and they answer much of the questions one would like to see answered about this work.

The writing is good. I especially enjoyed reading the Introduction and Formal Setup.

**Weaknesses:**

- The most glaring flaw is that the paper gives its setup as being motivated by requiring agents' actions to be justifiable to humans, but the actual methodology is a few steps removed from humans: both in the training (the synthetic judge is *not* trained on a dataset of humans ranking actions by their justifiability, but instead on a dataset of ground-truth clinician actions) and in evaluation (the evaluations measure the synthetic judge's judgments, instead of human judgments). I feel that the paper would be stronger with at least human evaluations, if not human preference data for training.
  - Also, the abstract says that "we showcase that agents trained via multi-agent debate learn to propose evidence that is resilient to refutations and *closely aligns with human preferences*." Which human preferences is this referring to? Does it just mean that the actions taken are similar to the actual clinicians' actions? Does this presuppose that clinicians' actions == justifiable actions? If so, is this stated anywhere?



Minor:

- The definition of the Preference Dataset (Section 4.1) seems strange: the preference indicates which of the two decisions is more justified by a given set of evidence in a particular state. But what is that set of evidence? Shouldn't that evidence be included in the tuple? Indeed, in the experiments, the preference dataset used does not fulfill this definition, since $p$ simply indicates the actions that the real clinicians took, not the action that they thought was more justifiable given the (sampled at random) evidence.


Nitpicks:
- Perhaps footnote 3 is important enough to not be a footnote. I was searching for a bit to figure out where $a^B_t$ came from in the experiments.

---

I think this paper isn't perfect but it's a good proof-of-concept of an interesting idea, so I recommend it for (borderline) acceptance.

**Questions:**

- In the abstract: the reward from the debate-based reward model yields "effective policies highly favored by the judge when compared to [the baseline]" -- the obvious question in the reader's mind is whether the "effective policies" actually yield higher environment reward than the baseline or not, and it would be nice to not have that ambiguity. If I'm reading the results correctly, they yielded lower performance, so maybe something like "while being nearly as effective as [the baseline]" or "while barely sacrificing any performance" would be nice.
- Shouldn't the description of the Baseline Agent (Section 3.1) indicate that the baseline agent is learned and therefore in practice the baseline agent is approximately optimal, but not necessarily exactly optimal?
- In the definition of the Justifiable Agent (Section 3.1), I think it would be helpful to explain what $a^B_t$ is. Even though it's described in the next section, it would clarify the dependence of $r^d$ and thus $R_J$ on the baseline agent's policy right away.
- Debate Game (Section 3.2): I'm just curious: Did you consider defining U to be the difference between the judge's judgments instead of the binary +1, -1? If you did, why did you choose not to use it?
- Using uniform random evidence to train the judge is surprising at first. Do you think it works well? Are there any alternatives? (can you keep training the judge in a loop while the argumentative agents train, perhaps?) An ideal dataset would actually include pairs of action + evidence, and human rankings over those, right?
- Creating the synthetic dataset by comparing a random action with the clinician's action seems like it would result in a lot of pairs in the dataset where one action is clearly superior to the other. Is this true? Does this cause any issues?
- In Section 5.4: "not only can agents significantly amplify the capabilities of a judge, but they also manage to recognize and correctly convince it of its true underlying preference" -- what are those two clauses referring to? I can't see how they're different.
- Section 5.4: Am I reading correctly that the isolated argumentative agent is trained with L = 6, yet in this setting it's only allowed to submit 3 evidences? How does that work (does the isolated one just pick 3 without regards to the choice of confuser, while the confuser picks 1 evidence after each isolated choice)? Wouldn't it be more correct to train it with L = 3?

---

> ### Author Response · Authors · 2023-11-16
> **Response to Reviewer 8w5y**
>
> Thank you for your insightful comments and questions. We are happy to see that you overall enjoyed the paper and that you consider the idea of using debate in this context an interesting research direction. In the following, we try to answer all of your questions and comments. The key points of our response can be summarized as follows:
> - We address your questions pertaining to the use of judge’s reward difference in the utility function, choice of an alternative action in the preference dataset and confuser evaluation of the isolated agent with additional experiments;
> - We clarify the definition of the preference dataset from Sec. 4.1, made assumption about the justifiable actions, sampling of evidence during training of the judge and approximate optimality of the baseline agent;
>
> ## Questions Involving Additional Experiments
>
> > Debate Game (Section 3.2): I'm just curious: Did you consider defining U to be the difference between the judge's judgments instead of the binary +1, -1? If you did, why did you choose not to use it?
>
> Thank you for the comment. We apologize for the confusion that may have occurred due to our mistake in writing of the utility function (Sec. 3.2). Throughout the paper, we assumed that when two actions are equally justifiable (e.g., as in a case when they are equal) the debate game draws, ending in both players obtaining a score of 0.
>
> Initially, we have only considered defining the debate utility function using rewards +1, 0, -1. However, the idea you put forward sounds interesting, and we examine it in more details. In particular, we have retrained maxmin and self-play agents with the utility function defined using a difference between predicted justifiability reward, namely $\\mathbb{U}(a_t, a_t^B, \\{e\\}) = \\mathcal{J}(a_t, \\{e\\}) - \\mathcal{J}(a_t^B, \\{e\\})$. In addition, we also train new confuser agents, so we can compare both effectiveness and robustness of the two approaches. We include the results in App. D.4, Plot 6c. The two approaches perform similarly in situations which do not involve a confuser agent. However, it seems that defining the utility using a difference in judge’s rewards leads to slightly lower scores when debate agents are faced with an adversary.
>
> > Creating the synthetic dataset by comparing a random action with the clinician's action seems like it would result in a lot of pairs in the dataset where one action is clearly superior to the other. Is this true? Does this cause any issues?
>
> Thank you for the question. We hypothesize that a situation where one action is clearly superior to the other would happen only in relatively rare cases (e.g., pairing an action of “no medication” with an action “highest IV/VC dose”, for a patient whose vitals are normal). In the context of our framework, the “difficulty” of the synthetic dataset primarily impacts the quality of a learned judge model. To empirically test this, we now examine a total of 3 different methods for pairing an alternative action:
> - **random**: true action $a_t$ is paired with an alternative action sampled uniform-random (as described in the main text);
> - **exhaustive**: true action $a_t$ is paired with every possible alternative action;
> - **offset**: true action $a_t$ is paired with an action in the neighborhood of $a_t$. To define a neighborhood, we recall that there are 5 choices for both, vasopressors (VC) and intravenous fluids (VC). Therefore, we can write $a_t = 5 * IV + VC$, where $IV, VC \\in \\{0, 1, 2, 3, 4\\}$. To obtain an alternative action, we consider changing IV and VC by an offset sampled uniform-random from a set $\\{-1, 0, 1\\}$, for both IV and VC. For example, for $a_t=21$ we have $IV=4$ and $VC=1$, and we randomly select an action that has $IV \\in \\{3, 4, 5\\}$ and $VC \\in \\{0, 1, 2\\}$ (ensuring we don’t end up with the same action).
>
> For each of the 3 variants, we train a new judge model using the same procedure described in the main paper, and show the accuracy on the test set in App D.5, Plot 6d. The exhaustive variant represents the most informative, but also unrealistically large, dataset. The random variant represents somewhat of a “middle ground” in terms of dataset difficulty. Lastly, the offset variant represents the most difficult case, as differences between two actions are more nuanced. However, while the achieved accuracies reflect the difficulty of the corresponding dataset (i.e., the highest accuracy for the exhaustive variant, followed by random and offset variants), the difference seems to be small enough that we can assume our choice of the random variant did not cause issues.

---

> > ### Author Response · Authors · 2023-11-16
> > **Response to Reviewer 8w5y (cont.)**
> >
> > > Section 5.4: Am I reading correctly that the isolated argumentative agent is trained with L = 6, yet in this setting it's only allowed to submit 3 evidences? How does that work (does the isolated one just pick 3 without regards to the choice of confuser, while the confuser picks 1 evidence after each isolated choice)? Wouldn't it be more correct to train it with L = 3?
> >
> > Thank you for raising this concern. The isolated argumentative agent was initially trained to propose L=6 evidence, whereas during evaluation against a confuser, it was proposing L=3 evidence in a debate-like setup. As a response to your comment, we introduce a new setup for the isolated agent (also described as part of the changes made to the paper in our general response). In the new version, the isolated agent is trained to propose L=3 evidence. When evaluating robustness, we do not perform debate, but rather let the agent first propose L=3 evidence, followed by a confuser proposing the remainder. We report the new results in the main text (Sec. 5.4, Plot 4b). We also include the previous setup and do a comparison to the updated one in App. C.2.1 and App. D.3. The conclusions we drew about robustness of the isolated agent in Sec. 5.4 remain the same in both cases.
> >
> > ## Discussion on Paper Weaknesses
> >
> > > The most glaring flaw is that the paper gives its setup as being motivated by requiring agents' actions to be justifiable to humans, but the actual methodology is a few steps removed from humans: both in the training (the synthetic judge is not trained on a dataset of humans ranking actions by their justifiability, but instead on a dataset of ground-truth clinician actions) and in evaluation (the evaluations measure the synthetic judge's judgments, instead of human judgments). I feel that the paper would be stronger with at least human evaluations, if not human preference data for training.
> >
> > Thank you for the comment. We acknowledge that our results are reported using a learned, proxy judge, without conducting further human evaluations. In this work, we have focused on a particular domain of treating sepsis and aimed at performing an in-depth examination of the intricacies of our method, providing a proof-of-concept exploring a particular approach to learning to justify. Future work could provide further human evaluations, strengthening the claims made in this paper, and tackle the challenges in collecting of human preference data, which we briefly touched upon in the discussion section.
> >
> > > Also, the abstract says that "we showcase that agents trained via multi-agent debate learn to propose evidence that is resilient to refutations and closely aligns with human preferences." Which human preferences is this referring to? Does it just mean that the actions taken are similar to the actual clinicians' actions? Does this presuppose that clinicians' actions == justifiable actions? If so, is this stated anywhere?
> >
> > Thank you for the question. This refers to the synthetic preference we designed in Sec. 5.1, which in turn presupposes that a clinician’s action is in fact a justifiable action. We have updated the paper to make this clearer to the reader.
> >
> > ## Response to General Comments and Questions
> >
> > > The definition of the Preference Dataset (Section 4.1) seems strange: the preference indicates which of the two decisions is more justified by a given set of evidence in a particular state. But what is that set of evidence? Shouldn't that evidence be included in the tuple? Indeed, in the experiments, the preference dataset used does not fulfill this definition, since simply indicates the actions that the real clinicians took, not the action that they thought was more justifiable given the (sampled at random) evidence.
> >
> > Thank you for noticing inconsistencies with our presentation here. We have mistakenly referred to the evidence used during training of a judge. The preference dataset D, as it was intended to be defined in the section 4.1, simply indicates which of the two actions was more preferred/justified in a given state. We have corrected the typo, and we apologize for any confusion this might have caused.
> >
> > > Perhaps footnote 3 is important enough to not be a footnote. I was searching for a bit to figure out where came from in the experiments.
> >
> > Thank you for the comment. We have revised the paper accordingly.

---

> > > ### Author Response · Authors · 2023-11-16
> > > **Response to Reviewer 8w5y (cont.)**
> > >
> > > > In the abstract: the reward from the debate-based reward model yields "effective policies highly favored by the judge when compared to [the baseline]" -- the obvious question in the reader's mind is whether the "effective policies" actually yield higher environment reward than the baseline or not, and it would be nice to not have that ambiguity. If I'm reading the results correctly, they yielded lower performance, so maybe something like "while being nearly as effective as [the baseline]" or "while barely sacrificing any performance" would be nice.
> > >
> > > Thank you for the comment. You have correctly interpreted the results, the justifiable policies indeed yield slightly lower performance, depending on the level of parameter lambda. We have made changes to the abstract so that this is better depicted.
> > >
> > > > Shouldn't the description of the Baseline Agent (Section 3.1) indicate that the baseline agent is learned and therefore in practice the baseline agent is approximately optimal, but not necessarily exactly optimal?
> > >
> > > We appreciate the comment. In practice, we only require the baseline policy to be well-performing, which avoids trivial solutions. For example, if we consider a case involving a poor quality baseline policy, it might be easy, but useless, to be more justifiable than it. We have updated the paper to better clarify this.
> > >
> > > > In the definition of the Justifiable Agent (Section 3.1), I think it would be helpful to explain what is. Even though it's described in the next section, it would clarify the dependence of  and thus on the baseline agent's policy right away.
> > >
> > > Thank you for the suggestion. We have revised the paper accordingly.
> > >
> > > > Using uniform random evidence to train the judge is surprising at first. Do you think it works well? Are there any alternatives? (can you keep training the judge in a loop while the argumentative agents train, perhaps?) An ideal dataset would actually include pairs of action + evidence, and human rankings over those, right?
> > >
> > > Thank you for the question. Our decision to train the judge using evidence sampled uniform-random stems from prior work [1], which used a conceptually similar approach (there, a sparse MNIST classifier). In addition, we believe that alternative approaches of sampling might lead to a biased judge model, which further corroborated our decision to stick with evidence sampled uniform-random. In practice, an ideal dataset would consist of triplets $(a_1, a_2, \{e\})$ paired with human rankings, which in turn would be used to train the judge model.
> > >
> > > > In Section 5.4: "not only can agents significantly amplify the capabilities of a judge, but they also manage to recognize and correctly convince it of its true underlying preference"  -- what are those two clauses referring to? I can't see how they're different.
> > >
> > > Thank you for noticing this. The two clauses refer to the same point: the agents amplify the capabilities of the judge. We have revised the paper accordingly.
> > >
> > > ## Conclusion
> > > We thank you again for your comments and questions. We are happy to answer anything else in addition.
> > >
> > > ## References
> > >
> > > 1. G. Irving, P. Christiano, and D. Amodei, “AI safety via debate.” arXiv:1805.00899

---

> ### Comment · Reviewer_8w5y · 2023-11-17
>
> Thank you for your very thorough response and the impressive amount of additional experiments performed. I already recommended the paper for acceptance, and with these corrections and improvements, the recommendation is only stronger.
>
> If it were possible, I would raise my rating from a 6 to a 7. Since I cannot, I will consider raising my rating to an 8 or keeping it at a 6.

---

> ### Author Response · Authors · 2023-11-18
> **Thank you for your comment**
>
> Thank you for your comment. We are pleased to see that you are positive about our work. We would be happy to respond to further questions, should you have any.

---

### Official Review · Reviewer_WJVW · 2023-11-06

**Soundness:** 3 good
**Presentation:** 2 fair
**Contribution:** 4 excellent
**Rating:** 8
**Confidence:** 3

**Summary:**

This paper considers the problem of designing RL agents that can provide evidence supporting their chosen actions ('justification') to a human observer - an important and timely problem that has not received enough attention. The authors propose an approach where a zero-sum debate extensive form game is played out between two agents, each attempting to convince a human judge (or proxy model) that their chosen action is most justified, by selecting 'evidence' (a subset of the state-space) to support the chosen action, contingent on the current state. The result of the human judge proxy model is a reward term which is mixed with the environmental reward, leading to an RL policy that trades off environmental reward and justifiability, as encoded by a dataset of human pairwise rankings of evidence proposals.

The paper is well written, with some minor grammatical errors. The proposed method is interesting and novel to my knowledge, and seems like a promising approach toward explainable RL policies. The empirical experiments are deep, but consider only one dataset - a medical problem of sepsis treatment. I have left some extensive comments and questions - but this is not a reflection of a weak paper, instead, this is due to my interest in the proposed method. The authors appear to be breaking new theoretical and empirical ground, and to this end the paper seems like a solid first effort in a promising direction.

I'm familiar with single-agent RL (specifically, my background is Inverse Reinforcement Learning), but might be unaware of some prior literature that would be relevant when performing my review. I read the main paper and appendices carefully. I have not carefully checked any theoretical derivations.

**Strengths:**

* A very important problem
 * A novel approach (debate-based justifications), and a solution method that seems promising
 * Clearly written, and the paper seems to represent high quality work
 * A compelling real-world empirical experimental case study
 * The comparison to SHAP as an alternative is great to see - and the proposed method fares well

**Weaknesses:**

* The method appears to require explicit knowledge of the optimal deterministic policy (called the baseline policy $\pi_B$ as part of the debate framework (unless I have misunderstood something). This seems like a big limitation in terms of practically applying this method to larger-scale problems.
 * Sec. 4.2 mentions that the judge proxy model follows a 'Bradley-Terry' model (and provides a citation). This seems like an important point in the design of your method, but I'm not familiar with this model or the background or history here. Can you elaborate on why this model is chosen?
 * The training of the justifiable agent (and the argumentative agent) appears to use offline i.i.d. sampling of (s, a, a) tuples (Sec 4.3). This seems like it could pose over/under-representation problems for some MDPs with deep trajectories that might be less or more common (i.e. problems where the i.i.d. assumption is more heavily violated by the dynamics). Can you comment about this limitation of the method?
 * Due to the additional structure of the debate game, the method introduces a number of additional hyper-parameters (e.g. evidence size $L$, mixing coefficient $\lambda$, scaling coefficient $\alpha$, as well as architecture of the various policy models, to say nothing of the learning algorithm hyper-parameters. Because of this, evaluation on one or more additional MDP problems would strengthen the evidence for the efficacy of this method.

**Questions:**

# Comments and questions

 * The use of an evidence-proposal debate setup for explainability is interesting and seems really promising to me. It seems like some deeper connections could be drawn to prior literature on the philosophical merits of debate as a form of explainability though. For instance, I'm curious about the following questions;
   * Why did you choose to have a multi-step debate framework, rather than a single step of evidence proposal? What is the interpretation or significance of the trajectory through the debate game tree (is there any?), or is the reached leaf node the only relevant outcome of the debate?
   * Similarly, how is a human RL designer or domain expert to interpret the proposal of multiple pieces of 'evidence' at each step of the debate game. If I understand correctly, evidence data are individual entries from the state-space vector (I think?). Is the interpretation of multiple pieces of evidence supposed to be a Boolean AND combination of each entry? (e.g. state feature A is present, AND B is present, therefore my action is better than the baseline policy)? I think a rich area for future exploration would be more structured evidence formats - e.g. "A is present, and C is present, but B is not present, therefore action X is chosen". Qualitative evaluations with domain experts would be beneficial in exploring this too.
 * There seem to be similarities between the debate-judge model and actor-critic RL approaches. Do you see any connections here? Could the proposed method be connected theoretically with A-C RL methods?
 * Similarly, as you note, the debate framework seems to have connections with pairwise preference ranking (e.g. Christiano et al. 2017). Given the popularity of RLHF at the moment, it might be illustrative to dig into the connections between this method and RLHF some more.
 * Fig 1. It appears that the baseline policy $\pi_B$ never acts in the environment - is this correct?
 * The paper refers to the debate-based reward approach as 'reward shaping' several times - please be careful with this term. Reward shaping refers technically to a specific subset of reward transformation that value functions are invariant to (see [A]). Although people often do abuse this term to refer to arbitrary designer interventions to alter environmental reward, it is best to be more careful with the terminology here to avoid confusion.
 * I note the judge reward $J \in \mathbb{R}$ is unbounded - this concerns me as a infinite range within which to rank items poses difficulties for a human judge (see psychology and human factors literature for example). Does your method still work with bounds on the judge reward? E.g. $J \in [-1, 1]$?
 * Can $\pi$ and $\pi_B$ output the same action and/or evidence? What happens in this situation? Is the judge proxy model able to output a 'no preference for either' choice?
 * Sec 4.4 mentions self-play is used during training. However I thought the agents being trained are deterministic. How can self-play be used in this case?
 * A compelling goal for future work would be the design of sample-efficient methods for specific MDPs that can operate with actual human-in-the loop judgements, rather than a proxy model.
 * The Sepsis reward design (App. B.3 is important - please provide references for the choice of $C_0, C_1, C_2$.
 * Given the specificity of your empirical experimental case-study, it would be helpful to have some more elaboration of the nature of the problem of sepsis control in the Appendix. I also assume the authors have some prior work or expertise or domain knowledge in this area. It would be beneficial to state if this is the case in the appendix to demonstrate that the research is being undertaken by an appropriately credentialed and/or experienced team (i.e. not computer scientists pretending to be doctors).

# Minor comments and grammatical points

 * Sec 3. - The MDP specification is missing a starting state distribution.
 * Sec 3.2 Why does $a_t$ act first in the debate game, not $a^B_t$. Did you test the reverse design?
 * You set the hyper-parameter $\alpha = 5$ in your experiments - why this value? Were other values tried?
 * Fig 3. caption - please define WIS so the reader doesn't have to refer to the main text.
 * App. B.2 - I believe you mean to say "setting the [log] probability of presented arguments to negative infinity".

# References

 * [A] Ng, Andrew Y., Daishi Harada, and Stuart Russell. "Policy invariance under reward transformations: Theory and application to reward shaping." Icml. Vol. 99. 1999.

---

> ### Author Response · Authors · 2023-11-16
> **Response to Reviewer WJVW**
>
> Thank you for your very perceptive comments and insightful questions. We are very pleased to see that you overall enjoyed the work and are happy to engage in further discussion. In the following, we try to answer all of your questions and comments. The key points of our response can be summarized as follows:
> - We address your concerns about requirement of the baseline policy and offline sampling during training;
> - We clarify our choices of the Bradley-Terry model, multi-step debate, and elaborate on a case when two actions are equally justifiable;
> - We discuss ways of interpreting proposed evidence, similarities to AC methods and connection to RLHF;
>
> ## Discussion on Paper Weaknesses
>
> > The method appears to require explicit knowledge of the optimal deterministic policy (called the baseline policy as part of the debate framework (unless I have misunderstood something). This seems like a big limitation in terms of practically applying this method to larger-scale problems.
>
> The method does require access to what we refer to as a baseline policy. However, this policy can be any well-performing policy, that is not necessarily optimal. The reason why we require a baseline policy to be well-performing is to avoid trivial solutions. In particular, if we consider a case involving a poor quality baseline policy, it might be easy, but useless, to be more justifiable than it. We have updated the paper to mention this.
>
> > Sec. 4.2 mentions that the judge proxy model follows a 'Bradley-Terry' model (and provides a citation). This seems like an important point in the design of your method, but I'm not familiar with this model or the background or history here. Can you elaborate on why this model is chosen?
>
> The choice of a Bradley-Terry model was inspired by the seminal work that examined specification of the reward function for reinforcement learning agents via preference ratings over trajectories [1]. It is a standard model for estimating score functions from pairwise preferences. In our framework, it can be understood in the context of equating judge’s rewards with a justifiability ranking scale. In particular, the difference in judge’s reward of two actions estimates the probability that a human considers one more justified than the other, given a particular set of evidence.
>
> > The training of the justifiable agent (and the argumentative agent) appears to use offline i.i.d. sampling of (s, a, a) tuples (Sec 4.3). This seems like it could pose over/under-representation problems for some MDPs with deep trajectories that might be less or more common (i.e. problems where the i.i.d. assumption is more heavily violated by the dynamics). Can you comment about this limitation of the method?
>
> Thank you for your comment. Regarding the argumentative agents (Sec. 4.3), it is important to acknowledge that in a general case of online RL there is a mismatch between a distribution of contexts used during their training (the i.i.d assumption), and a distribution of contexts that arises during the training of the justifiable agent. This distributional mismatch might affect the quality of the justifiable policy, since $\hat{r}^d$ might not approximate well $r^d$ in states that are much more frequently visited by the justifiable policy than by the behavior policy that generated the offline data. In our case, however, because we are operating in an offline reinforcement learning setup, the dataset used to train both argumentative and justifiable agents is the same, so we do not encounter such a distributional mismatch (i.e., states are sampled i.i.d during training of both agents). Regarding the justifiable agent, we believe that the limitations one could face in this setup are the same as those encountered in standard offline reinforcement learning problems [3]. Therefore, one could tackle out-of-distribution challenges using existing offline RL approaches, such as batch-constrained Q-learning [4] or Conservative Q-Learning [5].
>
> > Due to the additional structure of the debate game, the method introduces a number of additional hyper-parameters (e.g. evidence size, mixing coefficient, scaling coefficient, as well as architecture of the various policy models, to say nothing of the learning algorithm hyper-parameters. Because of this, evaluation on one or more additional MDP problems would strengthen the evidence for the efficacy of this method.
>
> Thank you for your comment. In this work, we focused on an in-depth analysis within the domain of treating sepsis, with a goal of achieving a comprehensive understanding of the intricacies of our method in a setting where learning to justify is of great importance. We agree that additional evaluation in different domains will provide further insights into the efficacy of the method, and it represents a part of the future work.

---

> > ### Author Response · Authors · 2023-11-16
> > **Response to Reviewer  WJVW (cont.)**
> >
> > ## Response to Comments and Questions
> >
> > > Why did you choose to have a multi-step debate framework, rather than a single step of evidence proposal? What is the interpretation or significance of the trajectory through the debate game tree (is there any?), or is the reached leaf node the only relevant outcome of the debate?
> >
> > The multi-step debate was chosen to best reflect a real-world argumentative procedure and is also based on the prior work [8]. The multi-step debate, or more specifically, a full path in the debate game tree, holds additional information that can be used to judge the outcome. From the perspective of a human, seeing the order of proposed evidence might give additional information about their relationship and mutual level of attack/defense that can be used to issue judgments [14]. Note that we have not fully exploited the structure of the multi-step debate in this work. In particular, the judge model is invariant to different paths that all lead to the same leaf node. Future work could consider approaches that can leverage the additional information provided by the multi-step debate framework.
> >
> > > Similarly, how is a human RL designer or domain expert to interpret the proposal of multiple pieces of 'evidence' at each step of the debate game. If I understand correctly, evidence data are individual entries from the state-space vector (I think?). Is the interpretation of multiple pieces of evidence supposed to be a Boolean AND combination of each entry? (e.g. state feature A is present, AND B is present, therefore my action is better than the baseline policy)? I think a rich area for future exploration would be more structured evidence formats - e.g. "A is present, and C is present, but B is not present, therefore action X is chosen". Qualitative evaluations with domain experts would be beneficial in exploring this too.
> >
> > Thank you for this insightful comment. In this work, the actual interpretation of logical relationships between proposed evidence is underspecified and left at the “discretion” of a judge. You are correct to say that the evidence are individual entries from the state-space vector, those most supportive of the decision to be justified. This represents all information the judge has to make a judgment. At the moment, argumentative agents are not capable of outputting rich relationships between evidence, in the form you suggested (e.g., “A” is present, but “B” is not, or “A” is present and makes the previously proposed evidence “C” obsolete, therefore “X” is chosen). There is a rich line of work that examines argumentation, both in social sciences [6] and in the domain of artificial intelligence [7], including argument definition, their mutual relations and ways of provably defending a particular claim. Like you already suggested, we see ample potential in combining our framework with symbolic AI approaches that would structurally enrich the debate with additional information that would both ease the judging of evidence and enable argumentative agents to better justify their claims.
> >
> > > There seem to be similarities between the debate-judge model and actor-critic RL approaches. Do you see any connections here? Could the proposed method be connected theoretically with A-C RL methods?
> >
> > Thank you for this very interesting comment. A potential connection we might see is in the existence of a judge model, which does act as a sort of critic: its judgments about justifiability of a made action are akin to “judgments” obtained from the critic in AC methods. However, while the two approaches might share similarities on a high level, there are notable technical differences. In particular, the critic in AC methods is based on return estimates, whereas in our framework, a judge model is used to directly assess the justifiability of an action in a given state.

---

> > > ### Author Response · Authors · 2023-11-16
> > > **Response to Reviewer  WJVW (cont.)**
> > >
> > > > Similarly, as you note, the debate framework seems to have connections with pairwise preference ranking (e.g. Christiano et al. 2017). Given the popularity of RLHF at the moment, it might be illustrative to dig into the connections between this method and RLHF some more.
> > >
> > > Thank you for your comment. We point out two notable connections to RLHF [1]-[2] that are worthy of investigation in the future work. First, in our work we perform a single training cycle: the judge, argumentative and justifiable agents are all trained once. It would be interesting to examine a procedure akin to RLHF, where we repeat the training procedure for multiple cycles, each time aiming in improving the justifiability of the policy obtained in the previous cycle. Second, we note that RLHF is generally limited to problems/domains where a human can effectively judge the outcome given a complete state/trajectory. Methods such as debate [8], recursive reward modeling [9] or iterated amplification [10] all aim in alleviating this restriction. Future work could also examine connection of the proposed method with RLHF in an attempt of scaling to domains where a human cannot fully comprehend the complete state/trajectory.
> > >
> > > > Fig 1. It appears that the baseline policy $\pi^B$ never acts in the environment - is this correct?
> > >
> > > This is correct, $\pi^B$ serves to provide the baseline action, but does not otherwise influence the dynamics of the environment. The justifiable agent is the sole acting entity in the MDP.
> > >
> > > > The paper refers to the debate-based reward approach as 'reward shaping' several times - please be careful with this term. Reward shaping refers technically to a specific subset of reward transformation that value functions are invariant to (see [A]). Although people often do abuse this term to refer to arbitrary designer interventions to alter environmental reward, it is best to be more careful with the terminology here to avoid confusion.
> > >
> > > Thank you for your comment. We acknowledge that we are slightly abusing the term “reward shaping”, as our “shaped” rewards do in fact induce learning of a new, justifiable policy, instead of preserving the original policy. We revised the paper to incorporate your feedback.
> > >
> > > > I note the judge reward J is unbounded - this concerns me as a infinite range within which to rank items poses difficulties for a human judge (see psychology and human factors literature for example). Does your method still work with bounds on the judge reward? E.g. J \in [-1, 1]?
> > >
> > > In our implementation, the judge is following the Bradley-Terry (BT) model, which in turn implies it is unbounded and real-valued. However, a human judge only gives a binary preference of the form “A is preferred to B” or “B is preferred to A” and does not otherwise specify any numerical value further describing this preference. The learned judge model we introduced in Sec. 4.2 is only a latent factor describing the human judgments, as estimated from the dataset.
> > >
> > > > Can $\pi$ and $\pi^B$ output the same action and/or evidence? What happens in this situation? Is the judge proxy model able to output a 'no preference for either' choice?
> > >
> > > Thank you for the comment. We apologize for the confusion that may have occurred due to our mistake in writing of the utility function (Sec. 3.2), as described in our general response. Throughout the paper, we assumed that when two actions are equally justifiable (e.g., as in a case when they are equal) the debate game draws, ending in both players obtaining a score of 0.
> > >
> > > > Sec 4.4 mentions self-play is used during training. However I thought the agents being trained are deterministic. How can self-play be used in this case?
> > >
> > > We use PPO to train the both, maxmin and self-play argumentative agents. During training, the argumentative policy is actually stochastic (i.e., we sample evidence from a categorical distribution defined by agent’s logits). During evaluation, we obtain a deterministic policy by taking the most likely action in a particular state (i.e., we perform the argmax operator over agent’s logits). We have updated the App. C.2 to describe this.
> > >
> > > > A compelling goal for future work would be the design of sample-efficient methods for specific MDPs that can operate with actual human-in-the loop judgements, rather than a proxy model.
> > >
> > > Thank you for the comment. We acknowledge that in this work we have not examined the sample-efficiency of the proposed framework. Future work could investigate the performance of argumentative agents trained with varying sizes of the preference dataset or, as you suggested, investigate approaches that can effectively operate with a human-in-the-loop.
> > >
> > > > The Sepsis reward design (App. B.3 is important - please provide references for the choice of C0, C1, C2.
> > >
> > > Thank you for noticing this. We have updated the paper to cite the relevant resource which was used to select the parameters.

---

> > > > ### Author Response · Authors · 2023-11-16
> > > > **Response to Reviewer WJVW (cont.)**
> > > >
> > > > > Given the specificity of your empirical experimental case-study, it would be helpful to have some more elaboration of the nature of the problem of sepsis control in the Appendix…
> > > >
> > > > Thank you for your comment. The problem of treating sepsis, in particular by applying reinforcement learning, has received significant attention in previous years [11]-[13], and it is on this line of work that we build upon. Because our goal was not to beat the state-of-art method for treating sepsis, we were very careful in making any premature conclusions about the performance of trained policies, apart from what has already been done in the previous work. We have additionally added a more detailed description of a general problem of sepsis treatment in App. B.1, in a hope of providing a better context and a necessary background for the reader.
> > > >
> > > > > Sec 3.2 Why does $a_t$ act first in the debate game, not  $a_t^B$. Did you test the reverse design?
> > > >
> > > > Actually, the starting player in the debate game is randomized, both during training and during evaluation. We briefly touch upon this in footnote 2. From the perspective of our framework, a starting player does not make a difference and only affects the order of evidence proposed.
> > > >
> > > > > You set the hyper-parameter $\alpha=5$  in your experiments - why this value? Were other values tried?
> > > >
> > > > The value for the parameter $\alpha$ was obtained through the hyperparameter tuning procedure. We have updated the paper to also include the range of values we tried, in this case [1, 5, 10, 15].
> > > >
> > > > > Minor comments and grammatical points
> > > >
> > > > Thank you for noticing inconsistencies in our presentation. We revised the paper to address your feedback.
> > > >
> > > > ## Conclusion
> > > >
> > > > Thank you again for your comments and questions. We are happy to answer anything else in addition.
> > > >
> > > > ## References
> > > >
> > > > 1. P. Christiano, J. Leike, T. B. Brown, M. Martic, S. Legg, and D. Amodei, “Deep reinforcement learning from human preferences.” arXiv:1706.03741
> > > > 2. L. Ouyang et al., “Training language models to follow instructions with human feedback.” arXiv:2203.02155
> > > > 3. S. Levine, A. Kumar, G. Tucker, J. Fu, “Offline Reinforcement Learning: Tutorial, Review, and Perspectives on Open Problems” arXiv:2005.01643
> > > > 4. S. Fujimoto, D. Meger, and D. Precup, “Off-Policy Deep Reinforcement Learning without Exploration.” arXiv:1812.02900
> > > > 5. A. Kumar, A. Zhou, G. Tucker, S. Levine, “Conservative Q-Learning for Offline Reinforcement Learning” arXiv:2006.04779
> > > > 6. F. H. Van Eemeren and R. Grootendorst, “Fundamentals of Argumentation Theory: A Handbook of Historical Backgrounds and Contemporary Developments,” College Composition and Communication, vol. 48, no. 3, p. 437, Oct. 1997.
> > > > 7. G. Simari and I. Rahwan, Eds., “Argumentation in Artificial Intelligence”, Boston, MA: Springer US, 2009.
> > > > 8. G. Irving, P. Christiano, and D. Amodei, “AI safety via debate.” arXiv:1805.00899
> > > > 9. J. Leike, D. Krueger, T. Everitt, M. Martic, V. Maini, and S. Legg, “Scalable agent alignment via reward modeling: a research direction.” arXiv:1811.07871
> > > > 10. P. Christiano, B. Shlegeris, and D. Amodei, “Supervising strong learners by amplifying weak experts.” arXiv:1810.08575
> > > > 11. M. Komorowski, L. A. Celi, O. Badawi, A. C. Gordon, and A. A. Faisal, “The Artificial Intelligence Clinician learns optimal treatment strategies for sepsis in intensive care,” Nat Med, vol. 24, no. 11, Art. no. 11, Nov. 2018.
> > > > 12. A. Raghu, M. Komorowski, I. Ahmed, L. Celi, P. Szolovits, and M. Ghassemi, “Deep Reinforcement Learning for Sepsis Treatment.” arXiv:1711.09602
> > > > 13. Y. Huang, R. Cao, and A. Rahmani, “Reinforcement Learning For Sepsis Treatment: A Continuous Action Space Solution” in Proceedings of the 7th Machine Learning for Healthcare Conference, PMLR, Dec. 2022.
> > > > 14. P. M. Dung, “On the acceptability of arguments and its fundamental role in nonmonotonic reasoning, logic programming and n-person games,” Artificial Intelligence, vol. 77, no. 2, pp. 321–357, Sep. 1995.

---

> > > > > ### Author Response · Authors · 2023-11-21
> > > > > **Closing of the Discussion Phase**
> > > > >
> > > > > Thank you once again for your feedback. As the rebuttal/discussion phase ends soon, we wanted to check if you had any additional comments. We hope that our response and the updates in the paper address your concerns. We would be happy to answer any further questions.

---

### Author Response · Authors · 2023-11-16
**General Response**

We sincerely thank the reviewers for detailed, attentive and careful reviews as well as insightful comments and questions. We are happy to see that reviewers find the overall problem important, that the proposed method is considered novel, and that the empirical results and performed experiments are compelling.  Apart from addressing each individual question in separate responses, we summarize the main changes in the paper, which address the concerns raised by the reviewers.

## Paper Changes

- In response to reviewer **WJVW**, we noticed a mistake in the writing of the utility function $\mathbb{U}$ (Sec. 3.2). In particular, throughout the paper we assumed that a case when two actions are equally justified yields a value of zero, but this was not correctly specified in writing. Therefore, we have updated the text to accurately depict this. The utility function returns $+1$ when $\\mathcal{J}(a_t, \\{ e \\}) > \\mathcal{J} (a_t^B, \\{ e \\})$, $0$ when $\\mathcal{J}(a_t, \\{ e \\}) = \\mathcal{J}(a^B_t, \\{ e \\})$, and $-1$ otherwise. We apologize for the confusion.
- In response to reviewer **2TSd**, we made changes to the presentation of Exp. 1 (Sec. 5.2, Plot 2b). In particular, we show percent of times actions proposed by the justifiable policy were preferred over those proposed by the baseline policy, but only when the two differ. Additionally, in App. D.2, we have included a plot that depicts the percent of times actions of the two policies were equally preferred, as well as the percent of times when one was more preferred than the other. This should provide further insights into differences between the two policies;
- In response to reviewers **8w5y** and **2TSd**, we updated the setup of Exp. 3 (Sec. 5.4) pertaining to the isolated agent. In particular, the agent is now trained to propose L=3 evidence, instead of L=6. When evaluating robustness (Sec. 5.4, Plot 4b), we do not perform debate, but rather let the isolated agent first propose its L=3 evidence, followed by a confuser, which proposes the remainder. We also include the previous setup and do a comparison to the updated one in App. C.2.1 and App. D.3. The conclusions we drew about robustness of the isolated agent in Sec. 5.4 remain the same in both cases;
- In response to reviewer **2TSd**, we have added a discussion on ethical considerations of our work;
- In response to reviewer **pkzb**, we have updated the related work section to include additional work on expert-in-loop systems;
- In response to general reviewer comments, we have rephrased and changed some sentences in the main text, which improved the clarity and readability of the paper. We have also updated the appendix with additional experiments.

### 21.11.2023

- We corrected a few typos, primarily in the appendices of the paper.

## Conclusion

We thank the reviewers once again for their reviews and are happy to answer any additional questions.

---

### Meta-Review · Area_Chair_fvPU · 2023-12-04

**Metareview:**

This paper investigates how to develop RL agents that take justifiable actions. To achieve this, the authors leverage a debate game that has two agents take turns providing evidence for two competing decisions, and have a judge to decide which is more justified.  With this debate game, it provides a reward term for justifiability that can be traded off with the environmental rewards to design justifiable policy. Extensive experiments on a single dataset have been performed to evaluate the performance of the proposed approach.

From a technical standpoint, there is consensus that this paper is solid. It tackles a well-motivated question with a novel and interesting approach, and the evaluations, though limited to one application, are convincing. However, concerns about the ethical implications of this approach have been raised, notably in the review by 2TSd.

We therefore recommend acceptance, conditional on proper ethical reviews. If accepted, we encourage the authors to provide a discussion on the ethical implications of the approach.

**Justification For Why Not Higher Score:**

While the paper is quite interesting and solid, given the scope focuses on one application and that the setup is sightly deviating from the original motivation, it is still more of a proof-of-concept.

**Justification For Why Not Lower Score:**

It addresses a well-motivated question, the proposed approach is novel, makes sense, and is interesting. The evaluations, while only on one application, are overall convincing.

---

### Decision · Program_Chairs · 2024-01-16

Accept (poster)